# Offline Actor-Critic for Average Reward MDPs

**William Powell**
Department of Mathematics
University of Wisconsin-Madison
Madison, WI 53706
`wgpowell@wisc.edu`

**Jeongyeol Kwon**
Wisconsin Institute for Discovery
Madison, WI 53706
`jeongyeol.kwon@wisc.edu`

**Qiaomin Xie**
Department of Industrial and Systems Engineering
University of Wisconsin-Madison
Madison, WI 53706
`qiaomin.xie@wisc.edu`

**Hanbaek Lyu**
Department of Mathematics
University of Wisconsin-Madison
Madison, WI 53706
`hlyu@math.wisc.edu`

## Abstract

We study offline policy optimization for infinite-horizon average-reward Markov decision processes (MDPs) with large or infinite state spaces. Specifically, we propose a pessimistic version of actor-critic methods using a computationally efficient linear function class for value function estimation. At the core of our method is a critic that computes a pessimistic estimate of the average reward under the current policy, as well as the corresponding policy gradient, by solving a fixed-point Bellman equation, rather than solving a successive sequence of regression problems as in finite horizon settings. Due to the nature of our policy-based method, the critic only needs to solve a linear optimization problem with convex quadratic constraints. We show that a very mild data coverage requirement is sufficient for our algorithm to achieve $O(\varepsilon^{-2})$ sample complexity for learning a near-optimal policy up to model misspecification errors. To our knowledge, this is the first result with optimal $\varepsilon$ dependence in the offline average reward setting.

## 1 Introduction

Reinforcement learning (RL) is a sequential decision making framework commonly studied in the online setting where an agent attempts to learn an optimal policy through active interactions with its environment. However, in many relevant applications such as autonomous driving and health care, online learning can be intractable or dangerous [Tang and Wiens, 2021]. In such cases, it is common to resort to *offline* learning, where the agent's goal is to learn a near-optimal policy from a static data set which was collected in a manner known to be safe and efficient for the application.

Without the ability to actively explore the environment, the agent's capability to learn is subject to the quality of the collected data. In particular, it is well known that effective learning is difficult or impossible if the data set does not sufficiently cover the space of states and actions. Thus, a major challenge in offline RL is the design of algorithms with provable guarantees under the weakest possible data coverage requirements. Initial work [Antos et al., 2007, Munos and Szepesvári, 2008] rely on strong uniform coverage assumptions that effectively require the data generating policy to visit the entire state-action space. However, this assumption is often unreasonable, especially for large or infinite state spaces. Accordingly, more recent work [Rashidinejad et al., 2021, Zanette et al., 2021, Zhan et al., 2022, Hong and Tewari, 2024, Li et al., 2024, Gabbianelli et al., 2024, Neu and Okolo, 2025] have focused on provable guarantees depending only on partial coverage conditions

39th Conference on Neural Information Processing Systems (NeurIPS 2025).

where the data set is only required to cover the subset of state-action pairs visited by an optimal policy.

State spaces in contemporary applications of RL can be very large or even infinite. Such applications necessitate the use of some form of function approximation for computational and memory efficient representation of policies and value functions. Empirical results show the promise of both simple linear models as well as more complex forms of function approximation such as neural networks, but there are still important theoretical gaps to be filled. This is especially true for infinite horizon MDPs, where value functions are solutions to a fixed point Bellman equation. As such, classical backward induction techniques that work well for episodic MDPs do not apply.

Our particular interest in this paper is offline RL for infinite horizon average reward MDPs (AMDPs) with linear function approximation. AMDPs are appropriate for continuing tasks such as inventory management [Giannoccaro and Pontrandolfo, 2002] or admission control [Weber et al., 2024], where there is no forced reset as in the episodic setting, or discounting that may lead to a myopic focus on short-term rewards. It is commonly acknowledged that theoretical analysis of algorithms for AMDPs is more challenging than that in the episodic or discounted setting. There are two primary reasons. First, as mentioned earlier, the techniques for episodic cases are not applicable here. Second, unlike discounted MDPs, the Bellman operator for AMDPs is not a contraction. Consequently, methods for discounted MDPs that rely on this contraction property do not carry over to the average reward setting.

In recent years, there have been a number of works investigating algorithms for AMDPs in the online setting [Fruit et al., 2018, Wei et al., 2020, 2021, Hao et al., 2021, Zhang and Xie, 2023, Agrawal and Agrawal, 2025, Hong et al., 2025], and learning from a generative model [Jin and Sidford, 2020, 2021, Zurek and Chen, 2024], as well as from deep RL and optimization perspectives [Zhang and Ross, 2021, Suttle et al., 2023, Agnihotri et al., 2024, Bai et al., 2024]. However, our understanding of the offline setting remains limited, especially with function approximation. The only work we know of is Primal-Dual Offline RL (PDOR) [Gabbianelli et al., 2024], which only achieves a sub-optimal $O(\varepsilon^{-4})$ sample complexity guarantee. Furthermore, PDOR's theoretical results require the strong assumption that the MDP's rewards and transition obey an exact linear structure, which rarely holds in practice. Motivated by this gap in the literature, the main question we ask in this work is the following:

> *Can we design a provably efficient algorithm for offline reinforcement learning in average reward MDPs with function approximation under minimal assumptions?*

We make strides towards an affirmative answer to this question by designing a pessimistic actor-critic algorithm using linear function approximation with a known feature map. Our key observation is that under our policy-based approach, a pessimistic estimate of the Bellman operator's fixed point can be computed by solving a simple linear optimization problem with convex quadratic constraints. We show our algorithm achieves optimal convergence rate guarantees with dependence on a measurement of data coverage used in prior work [Zanette et al., 2021, Gabbianelli et al., 2024, Neu and Okolo, 2025], which is referred to as the *feature coverage ratio*. This quantity is a measurement of how well the dataset aligns with the expected feature vector when following an optimal policy $\pi^*$. Our result's dependence on the intrinsic quantities improves over on the best known result for both average reward *and* discounted MDPs [Neu and Okolo, 2025]. Importantly, *we do not require the MDP itself to obey any linear structure*; our results hold under the more general conditions of approximate realizability and Bellman closedness. Furthermore, we only require the transition dynamics to satisfy a mild requirement that is significantly weaker than the uniform ergodicity assumption commonly considered for AMDPs [Wei et al., 2020, 2021, Bai et al., 2024]. Under these conditions, we show that $\tilde{O}(\varepsilon^{-2})$ samples are sufficient to learn a policy which is $\varepsilon$-optimal up to a model misspecification error.

## 1.1 Additional Related Work

Aside from the previously mentioned work [Zanette et al., 2021], another paper [Jin et al., 2021] also studies offline RL with linear function approximation in episodic MDPs. Their algorithm is a form of pessimistic least-squares value iteration, where pessimism is enforced through an additive bonus as commonly adopted in the online setting. It is not clear, however, whether this form of value iteration can be generalized to infinite horizon MDPs for the reasons discussed in the introduction.

Another line of related work studies offline RL in discounted MDPs with partial coverage and general function approximation [Xie et al., 2021, Cheng et al., 2022]. They also consider approximate realizability and Bellman closedness conditions similar to ours. The information theoretic results by [Xie et al., 2021] are near optimal, but due to their generality, theoretically optimal implementation of these algorithms is intractable. A computationally efficient alternative is presented in [Xie et al., 2021], but with sub-optimal convergence guarantees even when specialized to the linear setting. Similarly, convergence rates from the work [Cheng et al., 2022] are only of order $N^{-1/3}$ for $N$ data samples.

Finally, most closely related to this work are the papers by [Zanette et al., 2021, Gabbianelli et al., 2024, Hong and Tewari, 2024, Neu and Okolo, 2025]. We provide a detailed comparison with these works in Section 5.

## 2 Notation

For any set $\mathcal{X}$, we let $\Delta(\mathcal{X})$ be the set of all probability distributions on $\mathcal{X}$. Given a state space $\mathcal{S}$, action space $\mathcal{A}$, transition kernel $P : \mathcal{S} \times \mathcal{A} \to \Delta(\mathcal{S})$, and (stationary) policy $\pi : \mathcal{S} \to \Delta(\mathcal{A})$, the notation $\mathbb{E}_s^\pi[\cdot]$ is the expectation with respect to a Markov chain with transition kernel $P^\pi(s, s') = \sum_a \pi(a|s)P(s'|s, a)$ conditioned on starting in state $s \in \mathcal{S}$. For a function $f \in \mathbb{R}^\mathcal{S}$, we also use the notation $P_{s,a}f$ as short hand for the conditional expectation $\mathbb{E}_{s' \sim P(\cdot|s,a)}[f(s')]$. For a policy $\pi$ and probability measure $\mu \in \Delta(\mathcal{S})$, $\mu \otimes \pi$ is the probability measure on $\mathcal{S} \times \mathcal{A}$ defined by $(\mu \otimes \pi)(s, a) = \mu(s)\pi(a|s)$. Finally, given a symmetric, positive definite matrix $A \in \mathbb{R}^{d \times d}$, $\|\mathbf{x}\|_A$ denotes the norm on $\mathbb{R}^d$ defined by $\sqrt{\mathbf{x}^\top A \mathbf{x}}$.

## 3 Preliminaries

### 3.1 Average Reward MDPs

Let $\mathcal{S}$ be a large or possibly infinite state space and $\mathcal{A}$ be a finite space of $A$ actions. We consider an infinite horizon average reward MDP $(\mathcal{S}, \mathcal{A}, P, r)$ with transition kernel $P : \mathcal{S} \times \mathcal{A} \to \Delta(\mathcal{S})$ and reward function $r : \mathcal{S} \times \mathcal{A} \to [0, 1]$. An agent's rule for decision making in the MDP is specified by a (stationary) policy $\pi : \mathcal{S} \to \Delta(\mathcal{A})$ that maps current states to distributions over actions. At each time step $t$, the agent observes state $s_t$, takes action $a_t \sim \pi(\cdot|s_t)$, receives reward $r(s_t, a_t)$, then transitions to state $s_{t+1} \sim P(\cdot|s_t, a_t)$. The agents goal is to find a policy maximizing the average reward, which is defined as follows:

$$J^\pi(s) := \lim_{T \to \infty} \frac{1}{T} \mathbb{E}_s^\pi \left[ \sum_{t=0}^{T-1} r(s_t, a_t) \right]. \tag{1}$$

For each policy $\pi$, we define its associated Bellman operator $T^\pi : \mathbb{R}^{\mathcal{S} \times \mathcal{A}} \to \mathbb{R}^{\mathcal{S} \times \mathcal{A}}$ as

$$T^\pi f(s, a) = r(s, a) + \mathbb{E}_{s' \sim P(\cdot|s,a), a' \sim \pi(\cdot|s')}[f(s, a)], \qquad \forall s \in \mathcal{S}, a \in \mathcal{A}.$$

We will consider the following assumption throughout the paper.

**Assumption 3.1.** *All (stationary) policies induce a Markov chain that contains a single recurrent class and possibly some transient states (unichain). This implies that $J^\pi(s)$ is a constant independent of the initial state. Furthermore, it implies that for each policy $\pi$ there exists a function $q^\pi : \mathcal{S} \times \mathcal{A} \to \mathbb{R}$, unique up to linear translations, satisfying the Bellman equation*

$$q^\pi(s, a) + J^\pi = T^\pi q^\pi(s, a), \qquad \forall s \in \mathcal{S}, a \in \mathcal{A}, \tag{2}$$

*which we will call the q-function. In this case, $v^\pi(s) = \mathbb{E}_{a \sim \pi(\cdot|s)}[q^\pi(s, a)]$ is called the value function. We also assume there exists a constant $c$ such that*

$$\sup_\pi \|q^\pi\|_{sp} \leq c,$$

*where $\|q^\pi\|_{sp} = \sup_{s,a} q^\pi(s, a) - \inf_{s,a} q^\pi(s, a)$ is the span semi-norm.*

Under Assumption 3.1, for each policy $\pi$, there exists a unique stationary measure $\mu^\pi$ satisfying the equation

$$\sum_s \mu^\pi(s) \sum_a \pi(a|s) P(s'|s,a) = \mu^\pi(s'),$$

and we can write

$$J^\pi = \mathbb{E}_{(s,a) \sim \mu^\pi \otimes \pi}[r(s,a)].$$

As will be seen in the sequel, it is crucial for our algorithm that $J^\pi$ is constant and Assumption 3.1 is sufficient to guarantee this. This assumption also allows us to reliably estimate a bounded q-function to (2), which is necessary for stability in the policy improvement step of our algorithm.

We note that Assumption 3.1 is stronger than the weakly communicating assumption often considered in the online learning literature [Jaksch et al., 2010, Fruit et al., 2018, Wei et al., 2021, Hong et al., 2025]. However, it is important to point out one crucial fact. This assumption does *not* imply exploratory conditions such as uniformly lower bounded stationary measures [Wei et al., 2020] or a uniformly excited features condition [Hao et al., 2021, Wei et al., 2021]. These exploratory conditions imply that to cover the states visited by any policy, it is necessary to cover the entire state space. For more details on this, we refer the reader to Appendix A where we provide a more thorough discussion on the implications of Assumption 3.1 and comparison with other average reward models from the literature.

## 3.2 Offline RL

In offline RL, the agent only has access to a pre-collected data set $\mathcal{D} = \{(s_i, a_i, r_i, s_i')\}_{i=1}^N$, where $r_i = r(s_i, a_i)$ and each $s_i'$ is sampled from the conditional distribution $P(\cdot|s_i, a_i)$ independently of everything else. We do not need any additional assumptions about the collection of the data in $\mathcal{D}$. As in [Zanette et al., 2021, Neu and Okolo, 2025], the data does not need to be generated from i.i.d. sampling for from following a fixed behavior policy.

## 3.3 Function Approximation

Function approximation is necessary for learning in MDPs with large state and action spaces, where tabular solution methods are intractable. For actor-critic methods, this typically means using one function class $\Pi$ to represent policies, and another class $\mathcal{F}$ for value function estimation. Effective learning in the MDP then becomes highly dependent on the ability to accurately represent the true value functions for a given policy $\pi \in \Pi$ using functions in $\mathcal{F}$. To quantify this ability, we introduce the following definition.

**Definition 3.2.** *Let $\mathcal{F} \subset \mathbb{R}^{\mathcal{S} \times \mathcal{A}}$ be a set of functions used for value function approximation and $\Pi \subset \{\pi : \mathcal{S} \to \Delta(\mathcal{A})\}$ a class of stationary policies.*

*(i) We say that $(\mathcal{F}, \Pi)$ satisfies the approximate realizability property with constant $\kappa_{\mathcal{F},\Pi}$ if*

$$\sup_{\pi \in \Pi} \inf_{g \in \mathcal{F}, |\lambda| \leq 1} \|g + \lambda - T^\pi g\|_\infty \leq \kappa_{\mathcal{F},\Pi}. \tag{3}$$

*(ii) The tuple $(\mathcal{F}, \Pi)$ is said to satisfy the Bellman-restricted closedness property with constant $\varepsilon_{\mathcal{F},\Pi}$ if*

$$\sup_{\pi \in \Pi, f \in \mathcal{F}, |\lambda| \leq 1} \inf_{g \in \mathcal{F}} \|g + \lambda - T^\pi f\|_\infty \leq \varepsilon_{\mathcal{F},\Pi}.$$

These definitions are average-reward-analogues of those for episodic MDPs [Zanette et al., 2021, Nguyen-Tang and Arora, 2023]. Similar notions appeared in discounted MDPs in Xie et al. [Xie et al., 2021], although our use of the $\ell_\infty$ norm is slightly stronger than their requirement.

The approximate realizability property states that $\mathcal{F}$ nearly contains $q^\pi$ for each policy $\pi \in \Pi$. If $\kappa_{\mathcal{F},\Pi} = 0$ then the infimum in (3) is attained by $(g, \lambda) = (q^\pi, J^\pi)$.

However, realizability alone is known to be insufficient for sample efficient learning [Wang et al., 2021]. Therefore, additional conditions are often required. Restricted closedness measures how well

we can perform regression using functions $g \in \mathcal{F}$ when the target is the function resulting from the application of $T^\pi$ to a function $f \in \mathcal{F}$ plus a reasonable estimate $\lambda$ of $J^\pi$. The addition of $\lambda$ here is a generalization inspired, in part, by the requirement for linear MDPs that the column span of the feature matrix contains the all-one vector [Wei et al., 2021, Gabbianelli et al., 2024], as well as the development of generalized advantage estimation for the average reward setting in [Zhang and Ross, 2021] which includes a monte-carlo estimate of $J^\pi$ as part of the regression target.

With these definitions, we can now introduce the function and policy classes considered for our algorithm.

**Assumption 3.3.** *We consider the use of a linear function class*

$$\mathcal{Q}(B_w) := \left\{ q(s,a) = \phi(s,a)^\top \boldsymbol{w} : \|\boldsymbol{w}\|_2 \leq B_w \right\}$$

*where $B_w$ is a user-defined parameter and $\phi : \mathcal{S} \times \mathcal{A} \to \mathbb{R}^d$ is a known $d$-dimensional feature map with $\|\phi(s,a)\|_2 \leq 1$. We also assume a softmax policy class*

$$\Pi := \left\{ \pi(a|s) = \frac{e^{\phi(s,a)^\top \boldsymbol{\theta}}}{\sum_{a'} e^{\phi(s,a')^\top \boldsymbol{\theta}}} : \boldsymbol{\theta} \in \mathbb{R}^d \right\}. \tag{4}$$

*We assume that $(\mathcal{Q}(B_w), \Pi)$ satisfies the approximate completeness and Bellman-restricted closedness properties with constants $\kappa_{\mathcal{Q}(B_w),\Pi}$ and $\varepsilon_{\mathcal{Q}(B_w),\Pi}$ respectively.*

This assumption is a generalization of the widely studied linear MDP model, which is first introduced for episodic MDPs [Jin et al., 2020] and adapted to the average reward setting [Wei et al., 2021, Hong et al., 2025, Gabbianelli et al., 2024] with the realizability and restricted closedness constants being zero. Note that if the value functions are unbounded, it is unreasonable to assume approximate realizability. One cannot expect the ability to approximate functions in an unbounded set up to uniform error with an bounded function class. This is why we need the bounded span requirement in Assumption 3.1. However, we do not require prior knowledge of the constant $c$ in Assumption 3.1 for our results. Our theoretical guarantees remain true for any choice of $B_w$ provided Assumptions 3.1 and 3.3 hold.

# 4 Algorithm Details

Our algorithm is a form of pessimistic actor-critic method for the average reward setting. At a high level, it works through an alternating scheme run for a total number of $K$ iterations. First, at iteration $k$, given $\pi_k$, the pessimistic critic first computes the smallest plausible average reward of $\pi_k$ within a confidence region determined by the dataset $\mathcal{D}$. Then the actor updates the policy to $\pi_{k+1}$ through a conservative policy improvement step.

## 4.1 Pessimistic Policy Evaluation

Before giving a more precise description, we start with a motivating discussion of a natural idea inspired by methods in the episodic setting, but which turns out not to work in our setting. Assume for a moment that we have completed $k$ iterations of the algorithm, and we have an estimate $J_k$ of the average reward $J^{\pi_k}$ and an estimate $\hat{v}_k$ of the value function for policy $\pi_k$. Then by the Bellman equation (2), it is natural to estimate the q-function $q^{\pi_k}$ by solving the ridge regression problem

$$\boldsymbol{w}_k \in \underset{\boldsymbol{w} \in \mathbb{R}^d}{\arg\min} \left[ \sum_{i=1}^N \left( \phi(s_i, a_i)^\top \boldsymbol{w} + J_k - r_i - \hat{v}_k(s_i') \right)^2 + \|\boldsymbol{w}\|_2^2 \right],$$

which has the closed form expression

$$\boldsymbol{w}_k = \hat{\Lambda}^{-1} \sum_{i=1}^N \phi(s_i, a_i) \left( r_i - J_k + \hat{v}_k(s_i') \right). \tag{5}$$

Here, $\hat{\Lambda}$ is the un-normalized empirical covariance matrix

$$\hat{\Lambda} = \sum_{i=1}^N \phi(s_i, a_i) \phi(s_i, a_i)^\top + I,$$

and $I$ is the $d$-dimensional identity matrix. Let $\hat{q}_k(s, a) = \phi(s, a)^\top \boldsymbol{w}_k$ be our estimate of $q^{\pi_k}$. This method is similar to the typical approach for episodic MDPs, where the value function of step $h + 1$ is estimated and then used as a regression target to compute the weight vector for step $h$.

There are two main issues with this approach, however, in our AMDP setting. The first issue is about estimating $J^{\pi_k}$ in AMDPs, which we will address shortly. The second issue, also shared with discounted MDPs, is that the Bellman equation in our case is a fixed point equation. Consequently, we cannot use backward induction techniques from episodic MDPs. Therefore, it is unclear how to construct the estimated value $\hat{v}_k$ if we haven't already obtained an estimated Q-function $\hat{q}_k$ without resorting to Monte-Carlo methods. However, as pointed out in prior work [Zanette et al., 2021], Monte-Carlo estimation is undesirable in the offline setting: using importance sampling weights to cancel the distribution mismatch requires some knowledge of the data generating distribution, which is not available in most offline settings.

One method to address the second problem is to use $\hat{v}_{k-1}$, i.e. a value function estimate from the previous iteration, as the regression target (e.g. as in the work [Moulin and Neu, 2023]). However, this incurs additional bias and results in an additional term in the sub-optimality guarantee that must be handled in the analysis. This is usually done by showing that the difference in value functions between consecutive policies is small due to the conservative policy update. However, it remains unclear how to address this issue for AMDPs without strengthening Assumption 3.1 to include, for example, a uniform mixing assumption.

We propose to bypass these additional complexities and directly solve for the fixed point equation. Since $\boldsymbol{w}_k$ parametrizes our q-function estimates, we should also have $\hat{v}_k(s) = \mathbb{E}_{a \sim \pi_k(\cdot|s)}[\phi(s, a)^\top \boldsymbol{w}_k]$. Therefore, we replace $\hat{v}_k(s'_i)$ in (5) with $\phi^{\pi_k}(s'_i)^\top \boldsymbol{w}_k$, where $\phi^{\pi_k}(s) = \mathbb{E}_{a \sim \pi_k(\cdot|s)}[\phi(s, a)]$. Inspired by the ideas of Zanette et al. [2021], we then add a perturbation $\boldsymbol{\xi} \in \mathbb{R}^d$ to the weight vector and solve the following optimization problem:

$$(\boldsymbol{w}_k, \boldsymbol{\xi}_k, J_k) \in \underset{\boldsymbol{w}, \boldsymbol{\xi} \in \mathbb{R}^d, J \in \mathbb{R}}{\arg\min} \quad J$$

$$\text{s.t.} \quad \boldsymbol{w} = \boldsymbol{\xi} + \hat{\Lambda}^{-1} \sum_{i=1}^N \phi(s_i, a_i) \left( r(s_i, a_i) - J + \phi^{\pi_k}(s'_i)^\top \boldsymbol{w} \right), \tag{6}$$

$$|J| \leq 1, \quad \|\boldsymbol{w}\|_2 \leq B_w, \quad \text{and} \quad \|\boldsymbol{\xi}\|_{\hat{\Lambda}} \leq \beta,$$

where $\beta$ is a parameter determined by $B_w$, $K$, $N$, $\kappa_{\mathcal{Q}(B_w),\Pi}$, $\varepsilon_{\mathcal{Q}(B_w),\Pi}$, and a confidence level $\delta \in (0, 1)$. Specifically,

$$\beta = C + (\kappa_{\mathcal{Q}(B_w),\Pi} + \varepsilon_{\mathcal{Q}(B_w),\Pi})\sqrt{N} \tag{7}$$

where

$$C = O\left(B_w \sqrt{d \log(KNB_w/\delta)}\right). \tag{8}$$

This parameter quantifies the uncertainty in the dataset and our knowledge of the true MDP. The addition of $\boldsymbol{\xi}$ is how pessimism is incorporated into the algorithm. The ellipsoid $\|\boldsymbol{\xi}\|_{\hat{\Lambda}} \leq \beta$ can be viewed as a confidence set which, with high probability, contains the error due to lack of knowledge of the true transition. In our analysis, we will show that $J_k$ is a nearly pessimistic estimate of $J^{\pi_k}$ up to misspecification error determined by the constant $\kappa_{\mathcal{Q}(B_w),\Pi}$. This approach is also similar to the FOPO algorithm [Wei et al., 2021] for the online setting, but with one crucial difference: because our algorithm is policy-based, the first constraint in (6) is based on the Bellman equation for a *fixed* policy rather than the Bellman optimality equation. Consequently, the feasible set in (6) is convex, being comprised of linear and convex quadratic constraints. Therefore, approximate solutions to this optimization problem, up to arbitrarily small error, can be computed in polynomial time with interior point methods [Nesterov and Nemirovskii, 1994]. This stands in stark contrast to the FOPO algorithm, where our linear constraint in (6) is replaced by the analogous but nonlinear constraint

$$\boldsymbol{w} = \boldsymbol{\xi} + \hat{\Lambda}^{-1} \sum_{i=1}^N \phi(s_i, a_i)(r(s_i, a_i) - J + \max_a\{\phi(s'_i, a)^\top \boldsymbol{w}\}).$$

The nonlinearity comes from the additional maximization operation. This results in a non-convex constraint set and an optimization problem to be solved at every iteration without a known efficient computation method.

Finally, we remark here that while a fully efficient implementation of our algorithm would involve only approximate solutions to (6), for simplicity we will assume that $(\boldsymbol{w}_k, \boldsymbol{\xi}_k, J_k)$ is an exact solution for the remainder of the paper. We refer the interested reader to Section B of the appendix where we briefly discuss error propagation for a fully efficient implementation of our algorithm where (6) is solved approximately at each step.

## 4.2 Policy Update

Once the weight vector $\boldsymbol{w}_k$ is computed, the policy parameter is updated via

$$\boldsymbol{\theta}_{k+1} \leftarrow \boldsymbol{\theta}_k + \eta \boldsymbol{w}_k$$

where $\eta = \sqrt{\frac{\log A}{B_w^2 K}}$ is a step-size. Due to the form of policy class (4), this is equivalent to the exponential weights update

$$\pi_{k+1}(a|s) \propto \pi_k(a|s) \exp\left(\eta \hat{q}_k(s,a)\right), \quad \text{where } \hat{q}_k(s,a) = \phi(s,a)^\top \boldsymbol{w}_k$$

which, in turn, is equivalent to one step of mirror ascent with KL divergence regularization. Once the algorithm is terminated, it returns the output policy $\pi_{\text{out}}$, which is a mixture policy defined the uniform random sampling of policies $\{\pi_1, \ldots, \pi_K\}$. The pseudo code of our algorithm is presented in Algorithm 1.

---

**Algorithm 1** Average Reward Actor-Critic

---

1: **Input:** $\mathcal{D}$ (dataset), $B_w$ (function class parameter), $\beta$ (uncertainty parameter), $\eta$ (stepsize)
2: Form empirical covariance : $\hat{\Lambda} \leftarrow I + \sum_{i=1}^N \phi(s_i, a_i)\phi(s_i, a_i)^\top$
3: **Initialize:** $\boldsymbol{\theta}_1 = \boldsymbol{0}$
4: **for** $k = 1, \ldots, K$ **do**
5:     Let $(\boldsymbol{w}_k, \boldsymbol{\xi}_k, J_k)$ solve (6)
6:     Update policy parameter: $\boldsymbol{\theta}_{k+1} \leftarrow \boldsymbol{\theta}_k + \eta \boldsymbol{w}_k$.
7: **end for**
8: **Output:** $\pi_{\text{out}} = \text{Unif}[\pi_1, \ldots, \pi_K]$.

---

## 5 Main Results

Let $\hat{\Lambda}_N = \frac{1}{N}\hat{\Lambda}$ be the normalized covariance matrix. For a given comparator policy $\pi$ with stationary measure $\mu^\pi$, let $\phi^{\mu^\pi} = \mathbb{E}_{(s,a)\sim\mu^\pi\otimes\pi}[\phi(s,a)] \in \mathbb{R}^d$. The sub-optimality of the mixture policy $\pi_{\text{out}}$ output by Algorithm 1 with respect to a comparator policy $\pi$ depends on the random constant

$$\|\phi^{\mu^\pi}\|_{\hat{\Lambda}_N^{-1}},$$

referred to as the *feature coverage ratio* in [Gabbianelli et al., 2024, Neu and Okolo, 2025]. It measures how well the dataset $\mathcal{D}$ covers the feature space visited by $\pi$. When the *expected* feature vector $\phi^{\mu^\pi}$ is aligned with the top eigenvector of the empirical covariance matrix $\hat{\Lambda}$, one should expect this constant to be small—this is the case when $\mathcal{D}$ consists primarily of state-action pairs who's feature vectors are closely aligned with those frequently visited when following policy $\pi$. Our converge ratio can be contrasted with that used in prior work by [Jin et al., 2021]

$$\mathbb{E}_{(s,a)\sim\mu^\pi\otimes\pi}[\|\phi(s,a)\|_{\hat{\Lambda}_N^{-1}}],$$

which is no smaller than ours by Jensen's inequality.

Our main result is stated in Theorem 5.1 below.

**Theorem 5.1.** *Fix any comparator policy $\pi$ with stationary measure $\mu^\pi$. If we set $\beta$ as in (7), then with probability at least $1 - 2\delta$, for any $K \geq \log A$, Algorithm 1 run for $K$ iterations with stepsize $\eta = \sqrt{\frac{\log A}{B_w^2 K}}$ outputs a policy $\pi_{\text{out}}$ satisfying*

$$J^\pi - J^{\pi_{out}} \leq \underbrace{\frac{2C}{\sqrt{N}}\|\phi^{\mu^\pi}\|_{\hat{\Lambda}_N^{-1}}}_{T_1:\text{Uncertainty}} + \underbrace{2B_w\sqrt{\frac{\log A}{K}}}_{T_2:\text{Optimization}} + \underbrace{(2\|\phi^{\mu^\pi}\|_{\hat{\Lambda}_N^{-1}} + 1)(\varepsilon_{\mathcal{Q}(B_w),\Pi} + \kappa_{\mathcal{Q}(B_w),\Pi})}_{T_3:\text{Misspecification}}. \quad (9)$$

The upper bound on the sub-optimality gap (9) consists of three terms. The first term $T_1$ represents the uncertainty in the dataset. It decays with the reciprocal of the square root of the dataset's size, but increases as the quality of the dataset with respect to the comparator policy degrades, as measured by the coverage ratio. The second term $T_2$ is the error due to optimization, which can be made small by increasing the number of iterations $K$. The final term $T_3$ is an irreducible error due to model misspecification. Note that the $T_3$ term also decreases as the quality of the dataset improves. Importantly, due to the definition of $C$ in (8), the bound depends only on the feature dimension $d$ rather than on the size of the state space.

As an application of Theorem 5.1, let us consider the optimal policy $\pi^*$ as the comparator policy $\pi$. If $\|\phi^{\mu^{\pi^*}}\|_{\hat{\Lambda}^{-1}}$ is bounded above by some constant $C_*$, then Theorem 5.1 implies $\tilde{O}(B_w^2 C_*^2 d\varepsilon^{-2})$ samples are sufficient to learn a policy which is $\varepsilon$-optimal up to model misspecification error. One well-studied special case of zero misspecification error is the linear MDP [Jin et al., 2020, Wei et al., 2021]. Under this assumption, combined with Assumption 3.1, we can choose $B_w$ large enough so that the function class $\mathcal{Q}(B_w)$ contains the true value functions with the knowledge of an upper bound on $c$. Specifically, if $B_w \geq O(c\sqrt{d})$, a straightforward adaptation of our analysis shows that Theorem 5.1 holds with no misspecification error term.

## 5.1 Comparison with prior work

Our work is inspired by the work on episodic setting [Zanette et al., 2021], particularly the idea of solving a constrained optimization problem at each step. This work is also the first to introduce the definition of coverage ratio that we adopt in this paper. However, despite the algorithmic similarities, the algorithm design and analysis in this work rely crucially on backwards induction methods that are only applicable to the episodic setting. Our work makes a significant contribution by extending the approach to the more challenging infinite horizon setting.

As mentioned in the introduction, the work [Gabbianelli et al., 2024] is the only paper we are aware of that studies offline RL for average reward MDPs with linear function approximation. However, their algorithm only attains $O(\varepsilon^{-4})$ sample complexity guarantees. The main reason for this is their algorithm's double loop structure. Their primal-dual formulation of the offline RL problem involves solving for the saddle point of a certain Lagrangian objective, which is done through multiple rounds of stochastic gradient ascent-descent. They assume that state action pairs in the data set are sampled i.i.d from a fixed distribution. Then in each outer loop of their algorithm, they use $O(\varepsilon^{-2})$ samples to solve a sub-problem nearly exactly. Since they need $O(\varepsilon^{-2})$ outer-loop iterations, this results in a total sample complexity of $O(\varepsilon^{-4})$. In contrast, in our algorithm the data needed to construct the optimization problem (6) only needs to be sampled once and is then re-used in each iteration. This avoids the inner-loop that uses additional samples. The data re-use creates an additional correlation between iterates which is dealt with in the analysis using covering arguments.

The primal-dual algorithm by [Hong and Tewari, 2024] guarantees $O(\varepsilon^{-2})$ sample complexity for discounted MDPs, but their results depend on a weaker definition of coverage. Using our notation, their algorithm requires an upper bound on $\|\phi^{\mu^\pi}\|_{\hat{\Lambda}_N^{-2}}^2$, which is assumed to be known. Finally, [Neu and Okolo, 2025] introduce another primal-dual style algorithm with an $O(\varepsilon^{-2})$ sample complexity guarantee for discounted MDPs. Their suboptimality bounds depend on $\|\phi^{\mu^\pi}\|_{\hat{\Lambda}_N^{-1}}^2$, which is the strongest result we know of in this setting for the discounted case. The authors mention that it would not be difficult to adapt their results to average reward MDPs using the ideas from the work [Gabbianelli et al., 2024], but no additional details are provided. Even so, our work still improves over theirs for two reasons. First, while they use the same definition of coverage ratio as ours, their sub-optimality bounds scale quadratically with this constant in contrast to our linear scaling. Second, their results, like prior work [Gabbianelli et al., 2024, Hong and Tewari, 2024], only cover the more restrictive class of linear MDPs, while we study the more general linear function approximation (cf. Assumption 3.3).

A further extension beyond linear models to more general function approximation would be an interesting future direction. In this case, it is not yet clear how one can efficiently construct confidence sets for the underlying parameters. Even so, as shown in [Xie et al., 2021, Cheng et al., 2022], it is still possible to implement the pessimism principle with general function approximation. The

main difficulty we see in adapting the methods from these papers to the average reward setting is the presence of the additional variable $J$.

# 6 Analysis

Let $\hat{q}_k(s,a) := \phi(s,a)^\top \boldsymbol{w}_k$ and $\hat{v}_k(s) := \mathbb{E}_{a \sim \pi_k(\cdot|s)}[\hat{q}_k(s,a)]$ denote the empirical estimates of $q^{\pi_k}$ and $v^{\pi_k}$, respectively, at the end of $k$-th iteration. Key to our analysis is the following lemma, which allows us to decompose the sub-optimality in Theorem 5.1 into its three main parts as shown in (9).

**Lemma 6.1.** *Fix policies* $\pi, \tilde{\pi} \in \Delta(\mathcal{A})$. *Let* $\hat{J}^{\tilde{\pi}}$ *be an estimate of the true average reward following policy* $\tilde{\pi}$, *and* $\hat{q}^{\tilde{\pi}} \in \mathbb{R}^{\mathcal{S} \times \mathcal{A}}$ *be an estimate of the true Q-function* $q^{\tilde{\pi}}$. *Then*

$$J^\pi - \hat{J}^{\tilde{\pi}} = \mathbb{E}_{s \sim \mu^\pi}\left[\sum_a (\pi(a|s) - \tilde{\pi}(a|s))\hat{q}^{\tilde{\pi}}(s,a)\right] + \mathbb{E}_{(s,a) \sim \mu^\pi \otimes \pi}\left[T^{\tilde{\pi}}\hat{q}^{\tilde{\pi}}(s,a) - \hat{J}^{\tilde{\pi}} - \hat{q}^{\tilde{\pi}}(s,a)\right].$$

Lemma 6.1 is analogous to the so-called *extended performance difference lemma*, which is commonly used in the analysis of optimistic policy optimization algorithms for the online episodic setting; see, for example, the work [Cai et al., 2020, Shani et al., 2020]. Below we break down the analysis into three main steps.

**Step 1: Pessimism.** Suppose for the moment that $\kappa_{\mathcal{Q}(B_w),\Pi} = 0$. Then by the definition (3) there would exist some $\boldsymbol{w}_k^*$ such that $\phi(s,a)^\top \boldsymbol{w}_k^*$ solves the Bellman equation for policy $\pi_k$, meaning that $\phi(s,a)^\top \boldsymbol{w}_k^* = q^{\pi_k}(s,a)$. If we can show that $(\boldsymbol{w}_k^*, \boldsymbol{\xi}_k^*, J^{\pi_k})$ is feasible for critic's optimization problem (6), we will have $J_k \leq J^{\pi_k}$ by the definition of $J_k$, which has the desired pessimism property. In general though, if $\kappa_{\mathcal{Q}(B_w),\Pi} > 0$, it may be impossible to find such a $\boldsymbol{w}_k^*$. Therefore, we instead define

$$(\boldsymbol{w}_k^*, J_k^*) \in \underset{\|\boldsymbol{w}\|_2 \leq B_w, |J| \leq 1}{\arg\min} \|\phi(\cdot,\cdot)^\top \boldsymbol{w} + J - T^{\pi_k}(\phi(\cdot,\cdot)^\top \boldsymbol{w})\|_\infty \tag{10}$$

as the best possible weight vector and estimate of $J^{\pi_k}$ that incurs at most $\kappa_{\mathcal{Q}(B_w),\Pi}$ error by definition. With the help of Lemma 6.1, we then show that $J_k \leq J_k^* \leq J^{\pi_k} + \kappa_{\mathcal{Q}(B_w),\Pi}$ holds with high probability. This result is summarized in the following lemma.

**Lemma 6.2.** *With probability at least* $1 - \delta$, *for each* $k \in [K]$ *there exists* $\boldsymbol{\xi}_k^* \in \mathbb{R}^d$ *such that* $(\boldsymbol{w}_k^*, \boldsymbol{\xi}_k^*, J_k^*)$ *is feasible for the optimization problem* (6). *As a consequence of the definition of* $J_k$,

$$J_k \leq J_k^* \leq J^{\pi_k} + \kappa_{\mathcal{Q}(B_w),\Pi}, \quad \forall k \in [K].$$

**Step 2: Bounding the estimation error.** The next step is to control the error in the estimates $\hat{q}_k$ and $J_k$. More specifically, in view of the second term on the right hand side of Lemma 6.1, we are interested in bounding

$$\left|\mathbb{E}_{(s,a) \sim \mu^\pi \otimes \pi}\left[\hat{q}_k(s,a) + J_k - T^{\pi_k}\hat{q}_k(s,a)\right]\right|, \tag{11}$$

where $\pi$ is some comparator policy. In a manner similar to step 1, we define

$$\bar{\boldsymbol{w}}_k \in \underset{\|\boldsymbol{w}\|_2 \leq B_w}{\arg\min} \|\phi(\cdot,\cdot)^\top \boldsymbol{w} + J_k - T^{\pi_k}\hat{q}_k\|_\infty,$$

which is the best possible regression parameter with target $J_k - T^{\pi_k}\hat{q}_k$. In this case, we have $\|\phi(\cdot,\cdot)^\top \bar{\boldsymbol{w}}_k + J_k - T^{\pi_k}\hat{q}_k\|_\infty \leq \varepsilon_{\mathcal{Q}(B_w),\Pi}$ by Assumption 3.3. If $\varepsilon_{\mathcal{Q}(B_w),\Pi} = 0$, then $\hat{q}_k(s,a) - \phi(s,a)^\top \bar{\boldsymbol{w}}_k$ is exactly equal to quantity inside the expectation in (11). In the general case, (11) can be bounded by

$$|\mathbb{E}_{(s,a) \sim \mu^\pi \otimes \pi}[\hat{q}_k(s,a) - \phi(s,a)^\top \bar{\boldsymbol{w}}_k]| + \varepsilon_{\mathcal{Q}(B_w),\Pi}.$$

A high probability upper bound on the first term above results in the following Lemma 6.3.

**Lemma 6.3.** *With probability at least* $1 - \delta$, *for all* $k \in [K]$ *and any policy* $\pi$,

$$|\mathbb{E}_{(s,a) \sim \mu^\pi \otimes \pi}[\hat{q}_k(s,a) + J_k - T^{\pi_k}\hat{q}_k(s,a)]| \leq 2\beta\|\phi^{\mu^\pi}\|_{\hat{\Lambda}^{-1}} + \varepsilon_{\mathcal{Q}(B_w),\Pi}.$$

**Step 3: Completing the proof.** Theorem 5.1 holds on the events of Lemmas 6.2 and 6.3, which are true simultaneously with probability at least $1 - 2\delta$. The policy output by Algorithm 1 is a mixture policy with average reward $J^{\pi_{\text{out}}} = \frac{1}{K} \sum_{k=1}^{K} J^{\pi_k}$. Combining Lemma 6.2 with Lemma 6.1 we can show

$$\frac{1}{K} \sum_{k=1}^{K} J^{\pi} - J^{\pi_k} \leq \frac{1}{K} \sum_{k=1}^{K} J^{\pi} - J_k + \kappa_{\mathcal{Q}(B_w), \Pi}$$

$$\leq \frac{1}{K} \sum_{k=1}^{K} \mathbb{E}_{s \sim \mu^{\pi}} \left[ \sum_a (\pi(a|s) - \pi_k(a|s)) \hat{q}_k(s, a) \right]$$

$$+ \frac{1}{K} \sum_{k=1}^{K} \mathbb{E}_{(s,a) \sim \mu^{\pi} \otimes \pi} \left[ T^{\pi_k} \hat{q}_k(s, a) - J_k - \hat{q}_k(s, a) \right] + \kappa_{\mathcal{Q}(B_w), \Pi}.$$

The first term above is bounded above by the optimization error, which is proved through the analysis of mirror descent. The proof is completed by using Lemma 6.3, the definition of $\beta$, and rescaling $\hat{\Lambda}^{-1} = \frac{1}{N} \hat{\Lambda}_N^{-1}$.

## 7 Conclusion

In this paper, we have introduced a pessimistic actor critic algorithm for offline learning in infinite horizon average reward MDPs with linear function approximation. Our results show that our algorithm is sample efficient, with provable guarantees under only partial data coverage. One limitation of our algorithm is that we require the MDP to be unichain and value functions to have uniformly bounded span. Aside from an extension to general function approximation, potential future work could also include weakening Assumption 3.1 to include all weakly communicating MDPs.

## Acknowledgments and Disclosure of Funding

We thank all reviewers for their helpful suggestions and comments. The research of WP was supported in part by NSF Award DMS-2023239. JK was partially funded by AFOSR/AFRL grant no. FA9550-18-1-0166. HL was partially supported by NSF Award DMS-2206296. QX was supported in part by NSF grants CNS-1955997, ECCS-2339794, and ECCS-2432546.

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

# A  Further Discussion of Assumption 3.1

Here we give a more detailed discussion of Assumption 3.1 and how it compares to other average reward MDP models used in the literature.

The weakest assumption frequently made is the requirement that the MDP be weakly communicating [Jaksch et al., 2010, Fruit et al., 2018, Wei et al., 2020, 2021, Zhang and Xie, 2023, Hong et al., 2025, Zurek and Chen, 2024]. Aside from [Zurek and Chen, 2024] who study learning with access to a generative model, all of the works cited above study *online* learning in weakly communicating MDPs. We are not yet aware of any papers studying offline learning in weakly communicating MDPs.

An MDP is weakly communicating if the state space can be divided into two classes. One class consists of states that are transient for every policy. The other, called the communicating class, consists of a set of states with the following property: for each pair of states $(s, s')$ there exists a policy $\pi$ such that $s'$ is reachable from $s$ when following $\pi$ [Puterman, 2005]. As shown in [Jaksch et al., 2010], the weakly communicating assumption is necessary for efficient online learning. The property of being weakly communicating is sufficient to guarantee the existence of a solution $(q^*, J^*)$ to the Bellman optimality equation

$$q^*(s, a) + J^* = r(s, a) + \mathbb{E}_{s' \sim P(\cdot|s,a)}[\max_a q^*(s', a)]$$

where $J^*$ is constant. However, this is not enough to guarantee that for a fixed policy $\pi$, $J^\pi(s)$ as defined in (1) is constant. In this case, solutions $q^\pi$ to the Bellman equation

$$q^\pi(s, a) + J^\pi(s) = T^\pi q^\pi(s, a)$$

may exist, but may only be unique modulo a subspace of dimension greater than one. This is because, in weakly communicating MDPs, there can be policies that induce Markov chains with multiple recurrence classes. As such, the chain's stationary measure is not unique: there is a different one for each recurrence class. Moreover, while $q^*$ is bounded, there is no guaranteed uniform upper bound on the size of $q^\pi$ as opposed to discounted and episodic MDPs where $\frac{1}{1-\gamma}$ or $H$ provide a natural upper bound. This seems to make learning with policy optimization style algorithms difficult. It is not clear how to accurately estimate $q^\pi(s, a)$ in the weakly communicating case, and lack of a clear upper bound can destabilize the algorithm.

At the other end of the spectrum, the strongest assumption made in the literature is uniform ergodicity. The "ergodicity" part of this assumption requires that each policy $\pi$ induce an irreducible, aperiodic Markov chain. This ensures that the stationary measure is unique for each policy and positive everywhere. The "uniform" part of this assumption means that that the worst case mixing time

$$t_{\mathrm{mix}} = \sup_\pi \inf\{t \geq 1 : \max_s \|(P^\pi)^t(s, \cdot) - \mu^\pi\|_{TV} < \tfrac{1}{4}\} \tag{12}$$

is finite and all stationary measures are uniformly bounded away from zero:

$$\inf_{\pi,s} \mu^\pi(s) \geq \sigma > 0. \tag{13}$$

When learning with linear function approximation, this second condition is often replaced with

$$\mathbb{E}_{(s,a) \sim \mu^\pi \otimes \pi}\left[\phi(s,a)\phi(s,a)^\top\right] \succeq \lambda I, \tag{14}$$

meaning the true covariance matrix for each policy is uniformly positive definite. This is referred to as a "uniformly excited features" assumption in [Wei et al., 2021, Hao et al., 2021].

In the online setting, uniform ergodicity has been assumed in e.g. [Abbasi-Yadkori et al., 2019, Wei et al., 2020, 2021, Bai et al., 2024]. Under this assumption, the Bellman equation is solved by

$$q^\pi(s, a) = \sum_{t=0}^\infty \mathbb{E}_{s,a}^\pi\left[r(s_t, a_t) - J^\pi\right] \tag{15}$$

and $|q^\pi(s, a)| \leq O(t_{\mathrm{mix}})$. This is convenient for online learning, especially with policy optimization, because the $q$ functions are bounded and (13) and (14) imply that each policy is self-exploratory.

In the offline setting, assumptions such as (13) and (14) would greatly weaken single policy coverage results such as our Theorem 5.1. As discussed in the main body, effectively covering any policy requires covering the entire state space in the case of (14) or the entire feature space in the case of (14).

Because of this, it is important for us to emphasize that our assumptions do *not* imply conditions such as (13) or (14). Our Assumption 3.1, which is the same assumption made by [Gabbianelli et al., 2024], can be thought of as being somewhere in the middle of the two extremes mentioned above. It is a stronger than weakly communicating, but weaker than uniform ergodicity. Indeed, the uniform mixing condition (12) by itself implies Assumption 3.1 but is not necessary. Another sufficient condition is the existence of a single state $\bar{s}$ visited in

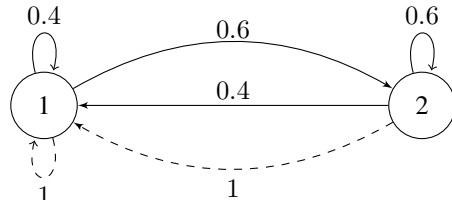

Figure 1: Two state river swim model. The dashed edges represent transition probabilities when taking action $L$ and the solid edges show the transition probabilities when taking action $R$.

expected time at most $h$ started from any other state as assumed in [Agrawal and Agrawal, 2025]. With this assumption, the span of the q-function is bounded by $h$.

We end this section with a simple example of an MDP which satisfies Assumption 3.1 but is not uniformly ergodic. Figure 1 diagrams a simple two state river-swim [Strehl and Littman, 2008] MDP where $\mathcal{S} = \{1, 2\}$ and $\mathcal{A} = \{L, R\}$. Conditioned on being in state 1, the agent stays in state 1 almost surely if action $L$ is chosen. If action $R$ is chosen, the agent will stay in state 1 with probability $0.4$ and transition to state 2 with probability $0.6$. On the other hand, if in state 2, the agent deterministically transitions to state 1 if action $L$ is chosen and will stay in state 2 with probability $0.6$ if action $R$ is chosen. Otherwise, it transitions to state 1. A simple linear algebra computation shows that the stationary measure is given by

$$[\mu^\pi(1), \mu^\pi(2)] = \left[ \frac{1 - 0.6\pi(R|2)}{1 + 0.6(\pi(R|1) - \pi(R|2))}, \frac{0.6\pi(R|1)}{1 + 0.6(\pi(R|1) - \pi(R|2))} \right].$$

for any $\pi$. If the agent never takes action $R$ in state 1, i.e. $\pi(R|1) = 0$, then $\mu^\pi(2) = 0$ so condition (13) fails. It is also easy to see that the expected covariance matrix, as in (14), is rank 1 and so cannot be positive definite if $d \geq 2$. However, it is not hard to show that for any initial distribution $\nu$ and policy $\pi$,

$$|(\nu P^\pi)^{t+1}(j) - \mu^\pi(j)| \leq 0.6|(\nu P^\pi)^t(j) - \mu^\pi(j)|$$

for $j \in \{1, 2\}$. This gives a upper bound on $t_{\text{mix}}$ of $\frac{5 \log 4}{2}$, which by (15) implies an upper bound on the span of $q^\pi$.

# B  A Fully Computationally Efficient Implementation

As discussed in Section 4, Algorithm 1 as written in the main body is not fully computationally efficient. This is because Line 5 assumes that the parameters $(\boldsymbol{w}_k, \boldsymbol{\xi}_k, J_k)$ are an exact solution to the optimization problem (6). It is not guaranteed that an exact solution to this problem can be computed efficiently. However, approximate solutions to (6) can be computed in polynomial time using interior point methods [Nesterov and Nemirovskii, 1994]. In this section, we briefly show how a fully computationally efficient implementation of Algorithm 1 where (6) is solved only approximately in step 5 results only in a small additive error proportional to the accuracy of the solution.

Specifically, for some tolerance parameter $\eta > 0$, let us assume that $(\boldsymbol{w}_k, \boldsymbol{\xi}_k, J_k)$ are a $\eta$-approximate solution to (6). This means that $J_k$ is within $\eta$ of the optimal solution and the magnitude of the constraint violation is at most $\eta$ when measured in the $\ell_2$ norm. Then the inequality in Lemma 6.2 may be updated to read

$$J_k \leq J_k^* + \eta \leq J^{\pi_k} + \kappa_{\mathcal{Q}(B_w), \Pi} + \eta.$$

The error in the constraint violation will appear in Lemma 6.3. If we approximately solve (6) so that $(\boldsymbol{w}_k, \boldsymbol{\xi}_k, J_k)$ satisfy

$$\left\| \boldsymbol{w}_k - \boldsymbol{\xi}_k - \hat{\Lambda}^{-1} \sum_{i=1}^N \phi(s_i, a_i) \left( r(s_i, a_i) - J + \phi^{\pi_k}(s_i')^\top \boldsymbol{w}_k \right) \right\|_2 \leq \eta,$$

$$|J_k| \leq 1 + \eta, \quad \|\boldsymbol{w}_k\|_2 \leq B_w + \eta, \quad \text{and} \ \|\boldsymbol{\xi}_k\|_{\hat{\Lambda}} \leq \beta + \eta,$$

then a quick inspection of the proof of Lemma 6.3 (more specifically Lemma C.3 below) shows that the result can be updated to read

$$|\mathbb{E}_{(s,a) \sim \mu^\pi \otimes \pi}[\hat{q}_k(s,a) + J_k - T^{\pi_k} \hat{q}_k(s,a)]| \leq 2(\beta + \eta)\|\phi^{\mu^\pi}\|_{\hat{\Lambda}^{-1}} + \varepsilon_{\mathcal{Q}(B_w), \Pi} + \eta.$$

Altogether, this only results an additional error of

$$2\eta \|\phi^{\mu^\pi}\|_{\hat{\Lambda}^{-1}} + 2\eta$$

added to the right hand side of (9) in Theorem 5.1.

# C   Proofs of Key Lemmas and Theorem 5.1

In this section, we will prove the key Lemmas from section 6 and also give a rigorous proof of Theorem 5.1. We start with Lemma 6.1.

**Proof of Lemma 6.1.** Define $\hat{v}^{\tilde{\pi}}(s) = \sum_a \tilde{\pi}(a|s)\hat{q}^{\tilde{\pi}}(s,a)$. Adding and subtracting $r(s,a) - \hat{J}^{\pi} + P_{s,a}\hat{v}^{\tilde{\pi}}$ and then using $J^{\pi} = \mathbb{E}_{(s,a)\sim\mu^{\pi}\otimes\pi}[r(s,a)]$ we have

$$\mathbb{E}_{s\sim\mu^{\pi}}\left[\sum_a \pi(a|s)\hat{q}^{\tilde{\pi}}(s,a)\right]$$

$$= \mathbb{E}_{s\sim\mu^{\pi}}\left[\sum_a \pi(a|s)\left(\hat{q}^{\tilde{\pi}}(s,a) + \hat{J}^{\tilde{\pi}} - r(s,a) - P_{s,a}\hat{v}^{\tilde{\pi}}\right)\right]$$

$$+ \mathbb{E}_{s\sim\mu^{\pi}}\left[\sum_a \pi(a|s)\left(r(s,a) - \hat{J}^{\tilde{\pi}} + P_{s,a}\hat{v}^{\tilde{\pi}}\right)\right]$$

$$= \mathbb{E}_{(s,a)\sim\mu^{\pi}\otimes\pi}\left[\hat{q}^{\tilde{\pi}}(s,a) + \hat{J}^{\tilde{\pi}} - T^{\tilde{\pi}}\hat{q}(s,a)\right] + J^{\pi} - \hat{J}^{\tilde{\pi}} + \mathbb{E}_{(s,a)\sim\mu^{\pi}\otimes\pi}\left[P_{s,a}\hat{v}^{\tilde{\pi}}\right]$$

Now, since $\mu^{\pi}$ is the stationary measure for policy $\pi$,

$$\mathbb{E}_{(s,a)\sim\mu^{\pi}\otimes\pi}\left[P_{s,a}\hat{v}^{\tilde{\pi}}\right] = \mathbb{E}_{s\sim\mu^{\pi}}[\hat{v}^{\tilde{\pi}}(s)] = \mathbb{E}_{s\sim\mu^{\pi}}\left[\sum_a \tilde{\pi}(a|s)\hat{q}^{\tilde{\pi}}(s,a)\right].$$

Inserting this into the last line above and re-arranging completes the proof.                                        $\square$

## C.1   Proofs of Lemmas 6.2 and 6.3

Before proving Lemmas 6.2 and 6.3, we need to introduce some additional notation and one auxillary result. Define the restricted policy class

$$\Pi(B_{\boldsymbol{\theta}}) = \left\{\frac{e^{\phi(s,a)^{\top}\boldsymbol{\theta}}}{\sum_{a'} e^{\phi(s,a')^{\top}\boldsymbol{\theta}}} : \|\boldsymbol{\theta}\|_2 \le B_{\boldsymbol{\theta}}\right\}$$

and the value function class

$$\mathcal{V}(B_{\boldsymbol{\theta}}, B_w) := \{v(s;\pi) = \langle\pi(\cdot|s), q(s,\cdot)\rangle : \pi \in \Pi(B_{\boldsymbol{\theta}}), q \in \mathcal{Q}(B_w)\}. \tag{16}$$

The number $B_{\boldsymbol{\theta}}$ represents the maximum size of any policy parameter $\boldsymbol{\theta}_k$ used during the course of $K$ iterations of Algorithm 1. Notice that, based on the policy update in Algorithm 1 we have

$$\|\boldsymbol{\theta}_k\|_2 \le \eta\sum_{k=1}^{K}\|\boldsymbol{w}_k\|_2 \le \eta K B_w$$

so we take $B_{\boldsymbol{\theta}} = \eta K B_w$.

The proofs of Lemmas 6.2 and 6.3 rely on the following uniform concentration inequality. Its proof is based on standard arguments from the literature first used in [Jin et al., 2020]. The proof of this lemma is delayed to Appendix D.

**Lemma C.1.** *Let $\{v_k\}_{k=1}^{K}$ be any, possibly random, collection from the function class $\mathcal{V}(B_{\boldsymbol{\theta}}, B_w)$. With probability at least $1 - \delta$ we have, for all $k \in [K]$,*

$$\left\|\sum_{i=1}^{N}\phi(s_i, a_i)\left(P_{s_i, a_i}v_k - v_k(s_i')\right)\right\|_{\hat{\Lambda}^{-1}}^2 \le \Gamma^2(B_w, N, \delta, K, d).$$

*where*

$$\Gamma^2(B_w, N, \delta, K, d) = 4B_w^2\left(\frac{d}{2}\log\left(\frac{K(N+1)}{\delta}\right) + d\log(1 + 4NB_w) + d\log(1 + 16NB_wB_{\boldsymbol{\theta}})\right) + 8.$$

We define $C$, the constant that appears in (8), as

$$C = \Gamma(B_w, N, \delta, K, d) + B_w.$$

The constant $\Gamma$ is derived from the well-known concentration of self-normalized processes (Lemma E.4) and the log-covering number of the class $\mathcal{V}(B_{\boldsymbol{\theta}}, B_w)$.

The next lemma shows that $J_k^*$ defined in (10) is within $\kappa_{\mathcal{Q}(B_w),\Pi}$ of $J^{\pi_k}$.

**Lemma C.2.** *Under Assumption 3.3 we have*

$$|J_k^* - J^{\pi_k}| \le \kappa_{\mathcal{Q}(B_w),\Pi}$$

*for every $k \in [K]$.*

*Proof.* Let $q_k^*(s,a) = \phi(s,a)^\top w_k^*$. By Lemma 6.1 and Assumption 3.3

$$
\begin{aligned}
|J^{\pi_k} - J_k^*| &= \left| \mathbb{E}_{(s,a) \sim \mu^{\pi_k} \otimes \pi_k} \left[ q_k^*(s,a) + J_k^* - T^{\pi_k} q_k^*(s,a) \right] \right| \\
&\le \| q_k^* + J_k^* - T^{\pi_k} q_k^* \|_\infty \\
&\le \kappa_{\mathcal{Q}(B_w),\Pi}.
\end{aligned}
$$

$\square$

We are now ready to prove Lemma 6.2. Throughout the proof, let

$$q_k^*(s,a) = \phi(s,a)^\top w_k^* \quad v_k^*(s) = \mathbb{E}_{a \sim \pi_k(\cdot|s)} \left[ q_k^*(s,a) \right]$$

be the approximate $q$ and value functions corresponding to the optimal parameter $w_k^*$.

***Proof of Lemma 6.2.*** By the definition (10), $w_k^*$ and $J_k^*$ are always feasible variables. Thus, we need to show that there is some $\boldsymbol{\xi}_k^* \in \mathbb{R}^d$ with $\|\boldsymbol{\xi}_k^*\|_{\hat{\Lambda}} \le \beta$ such that

$$w_k^* = \boldsymbol{\xi}_k^* + \hat{\Lambda}^{-1} \sum_{i=1}^N \phi(s_i,a_i) \left( r(s_i,a_i) - J_k^* + \phi^{\pi_k}(s_i')^\top w_k^* \right).$$

To this end, note that

$$
\begin{aligned}
w_k^* &= \hat{\Lambda}^{-1} \hat{\Lambda} w_k^* \\
&= \hat{\Lambda}^{-1} \sum_{i=1}^N \phi(s_i,a_i) \phi(s_i,a_i)^\top w_k^* + \hat{\Lambda}^{-1} w_k^* \\
&= \hat{\Lambda}^{-1} \sum_{i=1}^N \phi(s_i,a_i) \left( T^{\pi_k} q_k^*(s,a) - J_k^* \right) + \hat{\Lambda}^{-1} \sum_{i=1}^N \phi(s_i,a_i) \Delta_k(s_i,a_i) + \hat{\Lambda}^{-1} w_k^*. \qquad (17)
\end{aligned}
$$

where we define

$$\Delta_k(s,a) = q_k^*(s,a) + J_k^* - T^{\pi_k} q_k^*(s,a).$$

Adding and subtracting $v_k^*(s_i')$ from the first term in (17) we have

$$
\begin{aligned}
&\hat{\Lambda}^{-1} \sum_{i=1}^N \phi(s_i,a_i) \left( T^{\pi_k} q_k^*(s_i,a_i) - J_k^* \right) \\
&= \hat{\Lambda}^{-1} \sum_{i=1}^N \phi(s_i,a_i) \left( r(s_i,a_i) - J_k^* + v_k^*(s_i') \right) + \hat{\Lambda}^{-1} \sum_{i=1}^N \phi(s_i,a_i) \left( P_{s_i,a_i} v_k^* - v_k^*(s_i') \right).
\end{aligned}
$$

Now $v_k^*(s_i') = \mathbb{E}_{a \sim \pi_k(\cdot|s_i')}[q_k^*(s_i',a)] = \phi^{\pi_k}(s_i')^\top w_k^*$. So, plugging this back into (17) we have

$$w_k^* = \boldsymbol{\xi}_k^* + \hat{\Lambda}^{-1} \sum_{i=1}^N \phi(s_i,a_i) \left( r(s_i,a_i) - J_k^* + \phi^{\pi_k}(s_i')^\top w_k^* \right)$$

where

$$\boldsymbol{\xi}_k^* = \hat{\Lambda}^{-1} \sum_{i=1}^N \phi(s_i,a_i)(P_{s_i,a_i} v_k^* - v_k^*(s_i')) + \hat{\Lambda}^{-1} \sum_{i=1}^N \phi(s_i,a_i) \Delta_k(s_i,a_i) + \hat{\Lambda}^{-1} w_k^*. \qquad (18)$$

To complete the proof, we only need to show that $\|\boldsymbol{\xi}_k^*\|_{\hat{\Lambda}} \le \beta$ holds with high probability. Since $v_k^* \in \mathcal{V}(B_w, B_\theta)$ for all $k \in [K]$, by Lemma C.1 we have

$$
\begin{aligned}
\left\| \hat{\Lambda}^{-1} \sum_{i=1}^N \phi(s_i,a_i)(P_{s_i,a_i} v_k^* - v_k^*(s_i')) \right\|_{\hat{\Lambda}} &= \left\| \sum_{i=1}^N \phi(s_i,a_i)(P_{s_i,a_i} v_k^* - v_k^*(s_i')) \right\|_{\hat{\Lambda}^{-1}} \\
&\le \Gamma(B_w, N, \delta, K, d)
\end{aligned}
$$

holds for all $k$ with probability at least $1 - \delta$. Next, using the projection bound, Lemma E.3, and $|\Delta_k(s_i, a_i)| \leq \kappa_{\mathcal{Q}(B_w),\Pi}$ implied by Assumption 3.3, we have

$$\left\| \hat{\Lambda}^{-1} \sum_{i=1}^{N} \phi(s_i, a_i) \Delta_k(s_i, a_i) \right\|_{\hat{\Lambda}} = \left\| \sum_{i=1}^{N} \phi(s_i, a_i) \Delta_k(s_i, a_i) \right\|_{\hat{\Lambda}^{-1}} \leq \kappa_{\mathcal{Q}(B_w),\Pi} \sqrt{N}.$$

Finally, since $\hat{\Lambda}^{-1} \preceq I$,

$$\|\hat{\Lambda}^{-1} \boldsymbol{w}_k^*\|_{\hat{\Lambda}} = \|\boldsymbol{w}_k^*\|_{\hat{\Lambda}^{-1}} \leq \|\boldsymbol{w}_k^*\|_2 \leq B_w.$$

Therefore, by (18) and the triangle inequality,

$$\|\boldsymbol{\xi}_k^*\|_{\hat{\Lambda}} \leq \Gamma(B_w, N, \delta, K, d) + \kappa_{\mathcal{Q}(B_w),\Pi} \sqrt{N} + B_w \leq \beta.$$

holds with probability at least $1 - \delta$. The proof is complete. $\qquad\square$

We now move on to the proof of Lemma 6.3. Recall

$$\hat{v}_k(s) = \mathbb{E}_{a \sim \pi_k(\cdot|s)}[\hat{q}_k(s, a)]$$

as the definition of the empirical value function estimated by the critic. Since $\hat{v}_k(s) = \phi^{\pi_k}(s)^\top \boldsymbol{w}_k$, it follows from the definition of $(J_k, \boldsymbol{w}_k, \boldsymbol{\xi}_k)$ as the solution to the optimization problem (6) that

$$\boldsymbol{w}_k = \boldsymbol{\xi}_k + \hat{\Lambda}^{-1} \sum_{i=1}^{N} \phi(s_i, a_i)(r(s_i, a_i) - J_k + \hat{v}_k(s_i')). \qquad (19)$$

We will need the following statement.

**Lemma C.3.** *With probability at least $1 - \delta$, for each $k \in [K]$ we have*

$$|\mathbb{E}_{(s,a) \sim \mu}[\hat{q}_k(s, a) - \phi(s, a)^\top \bar{\boldsymbol{w}}_k]| \leq 2\beta \|\phi^\mu\|_{\hat{\Lambda}^{-1}}$$

*for any probability measure $\mu \in \Delta(\mathcal{S} \times \mathcal{A})$.*

*Proof.* For this proof, we define

$$\bar{\Delta}_k(s, a) = \phi(s, a)^\top \bar{\boldsymbol{w}}_k + J_k - T^{\pi_k} \hat{q}_k(s, a).$$

We observe that

$$\bar{\boldsymbol{w}}_k = \hat{\Lambda}^{-1} \hat{\Lambda} \bar{\boldsymbol{w}}_k$$

$$= \hat{\Lambda}^{-1} \sum_{i=1}^{N} \phi(s_i, a_i) \phi(s_i, a_i)^\top \bar{\boldsymbol{w}}_k + \hat{\Lambda}^{-1} \bar{\boldsymbol{w}}_k$$

$$= \hat{\Lambda}^{-1} \sum_{i=1}^{N} \phi(s_i, a_i)(T^{\pi_k} \hat{q}_k(s_i, a_i) - J_k) + \hat{\Lambda}^{-1} \sum_{i=1}^{N} \phi(s_i, a_i) \bar{\Delta}_k(s_i, a_i) + \hat{\Lambda}^{-1} \bar{\boldsymbol{w}}_k.$$

Using (19), this allows us to write the difference in the parameters $\boldsymbol{w}_k$ and $\bar{\boldsymbol{w}}_k$ as

$$\boldsymbol{w}_k - \bar{\boldsymbol{w}}_k = \boldsymbol{\xi}_k + \hat{\Lambda}^{-1} \sum_{i=1}^{N} \phi(s_i, a_i) \left( r(s_i, a_i) - J_k + \hat{v}_k(s_i') \right) - \hat{\Lambda}^{-1} \sum_{i=1}^{N} \phi(s_i, a_i)(T^{\pi_k} \hat{q}_k(s_i, a_i) - J_k)$$

$$- \hat{\Lambda}^{-1} \sum_{i=1}^{N} \phi(s_i, a_i) \bar{\Delta}_k(s_i, a_i) - \hat{\Lambda}^{-1} \bar{\boldsymbol{w}}_k$$

$$= \boldsymbol{\xi}_k + \hat{\Lambda}^{-1} \sum_{i=1}^{N} \phi(s_i, a_i) \left( \hat{v}_k(s_i') - P_{s_i a_i} \hat{v}_k \right) - \hat{\Lambda}^{-1} \sum_{i=1}^{N} \phi(s_i, a_i) \bar{\Delta}_k(s_i, a_i) - \hat{\Lambda}^{-1} \bar{\boldsymbol{w}}_k.$$

So we have

$$\phi(s, a)^\top (\boldsymbol{w}_k - \bar{\boldsymbol{w}}_k) = \phi(s, a)^\top \boldsymbol{\xi}_k + \phi(s, a)^\top \hat{\Lambda}^{-1} \sum_{i=1}^{N} \phi(s_i, a_i)(\hat{v}_k(s_i') - P_{s_i, a_i} \hat{v}_k)$$

$$- \phi(s, a)^\top \hat{\Lambda}^{-1} \sum_{i=1}^{N} \phi(s_i, a_i) \bar{\Delta}_k(s_i, a_i) - \phi(s, a)^\top \hat{\Lambda}^{-1} \bar{\boldsymbol{w}}_k.$$

Denote $\phi^\mu = \mathbb{E}_{(s,a)\sim\mu}[\phi(s,a)]$. Taking an expectation followed by an absolute value we get

$$
|\mathbb{E}_{(s,a)\sim\mu}[\hat{q}_k(s,a) - \phi(s,a)^\top \bar{w}_k]| = |(\phi^\mu)^\top(w_k - \bar{w}_k)|
$$
$$
\leq |(\phi^\mu)^\top \boldsymbol{\xi}_k| + \left|(\phi^\mu)^\top \hat{\Lambda}^{-1} \sum_{i=1}^N \phi(s_i,a_i)(\hat{v}_k(s_i') - P_{s_i,a_i}\hat{v}_k)\right|
$$
$$
+ \left|(\phi^\mu)^\top \hat{\Lambda}^{-1} \sum_{i=1}^N \phi(s_i,a_i)\bar{\Delta}_k(s_i,a_i)\right| + |(\phi^\mu)^\top \hat{\Lambda}^{-1}\bar{w}_k|. \tag{20}
$$

Let us consider each of the above terms on the right hand side. First, we have by Holder's inequality

$$
|(\phi^\mu)^\top \xi_k| \leq \|\phi^\mu\|_{\hat{\Lambda}^{-1}} \|\boldsymbol{\xi}_k\|_{\hat{\Lambda}} \leq \beta \|\phi^\mu\|_{\hat{\Lambda}^{-1}}
$$

where the second inequality is due to the constraint $\|\boldsymbol{\xi}\|_{\hat{\Lambda}} \leq \beta$ in the optimization problem (6). For the second term, using $\hat{v}_k \in \mathcal{V}(B_w, B_\theta)$ for all $k$, we appeal to Lemma C.1 to get

$$
\left|(\phi^\mu)^\top \hat{\Lambda}^{-1} \sum_{i=1}^N \phi(s_i,a_i)(\hat{v}_k(s_i') - P_{s_i,a_i}\hat{v}_k)\right| \leq \|\phi^\mu\|_{\hat{\Lambda}^{-1}} \left\|\sum_{i=1}^N \phi(s_i,a_i)(\hat{v}_k(s_i') - P_{s_i,a_i}\hat{v}_k)\right\|_{\hat{\Lambda}^{-1}}
$$
$$
\leq \Gamma(B_w, N, \delta, K, d)\|\phi^\mu\|_{\hat{\Lambda}^{-1}}
$$

with probability at least $1 - \delta$ for all $k$. For the third term, we use $|\bar{\Delta}_k(s,a)| \leq \varepsilon_{\mathcal{Q}(B_w),\Pi}$ by Assumption 3.3 and Lemma E.3 to get

$$
\left|(\phi^\mu)^\top \hat{\Lambda}^{-1} \sum_{i=1}^N \phi(s_i,a_i)\bar{\Delta}_k(s_i,a_i)\right| \leq \|\phi^\mu\|_{\hat{\Lambda}^{-1}} \left\|\sum_{i=1}^N \phi(s_i,a_i)\bar{\Delta}_k(s_i,a_i)\right\|_{\hat{\Lambda}^{-1}}
$$
$$
\leq \varepsilon_{\mathcal{Q}(B_w),\Pi}\sqrt{N}\|\phi^\mu\|_{\hat{\Lambda}^{-1}}.
$$

Finally, for the last term on the right hand side of (20) we simply observe that

$$
|(\phi^\mu)^\top \hat{\Lambda}^{-1}\bar{w}_k| \leq \|\phi^\mu\|_{\hat{\Lambda}^{-1}}\|\bar{w}_k\|_{\hat{\Lambda}^{-1}} \leq \|\phi^\mu\|_{\hat{\Lambda}^{-1}}\|\bar{w}_k\|_2 \leq B_w\|\phi^\mu\|_{\hat{\Lambda}^{-1}}
$$

Combining everything shows that with probability at least $1 - \delta$,

$$
|\mathbb{E}_{(s,a)\sim\mu}[\hat{q}_k(s,a) - \phi(s,a)^\top \bar{w}_k]| \leq \left(\beta + \Gamma(B_w, B, \delta, K, d) + \varepsilon_{\mathcal{Q}(B_w),\Pi}\sqrt{N} + B_w\right)\|\phi^\mu\|_{\hat{\Lambda}^{-1}}
$$
$$
\leq 2\beta\|\phi^\mu\|_{\hat{\Lambda}^{-1}}
$$

as desired. $\qquad\square$

***Proof of Lemma 6.3.*** Adding and subtracting $\phi(s,a)^\top \bar{w}_k$, we can decompose the Bellman error into two terms as

$$
\hat{q}_k(s,a) + \hat{J}_k - T^{\pi_k}\hat{q}_k(s,a) = \hat{q}_k(s,a) - \phi(s,a)^\top \bar{w}_k + \phi(s,a)^\top \bar{w}_k + \hat{J}_k - T^{\pi_k}\hat{q}_k(s,a).
$$

Therefore, by Lemma C.3 and the definition of $\bar{w}_k$,

$$
|\mathbb{E}_{(s,a)\sim\mu^\pi\otimes\pi}[\hat{q}_k(s,a) + \hat{J}_k - T^{\pi_k}\hat{q}_k(s,a)]| \leq |\mathbb{E}_{(s,a)\sim\mu^\pi\otimes\pi}[\hat{q}_k(s,a) - \phi(s,a)^\top \bar{w}_k]|
$$
$$
+ \|\phi(\cdot,\cdot)^\top \bar{w}_k + \hat{J}_k - T^{\pi_k}\hat{q}_k\|_\infty
$$
$$
\leq 2\beta\|\phi^{\mu^\pi}\|_{\hat{\Lambda}^{-1}} + \varepsilon_{\mathcal{Q}(B_w),\Pi}.
$$

$\qquad\square$

## C.2 Proof of Theorem 5.1

We are now ready to prove Theorem 5.1. We will need the following Lemma analyzing the actor's optimization procedure.

**Lemma C.4.** *Fix a comparator policy $\pi \in \Delta(\mathcal{A})$. We have*

$$
\mathbb{E}_{s\sim\mu^\pi}\left[\frac{1}{K}\sum_{k=1}^K \sum_a (\pi(a|s) - \pi_k(a|s))\hat{q}_k(s,a)\right] \leq 2B_w\sqrt{\frac{\log A}{K}}.
$$

*Proof.* Notice that the policy parameter update in Line 7 of Algorithm 1 can be written as the exponential weight update

$$
\pi_{k+1}(a|s) \propto \pi_k(a|s)\exp\left(\eta\hat{q}_k(s,a)\right).
$$

We have $|\hat{q}_k(s,a)| = |\phi(a,s)^\top \boldsymbol{w}_k| \leq B_w$ by Algorithm 1's constraint on $\boldsymbol{w}_k$ and the normalization of $\phi(s,a)$ in Assumption 3.3. We then appeal to standard analysis of exponential weights (Lemma E.1). If we choose $\eta \leq \frac{1}{B_w}$ then the conditions of the Lemma E.1 are satisfied and we have

$$\sum_{k=1}^{K}(\pi(a|s) - \pi_k(a|s))\hat{q}_k(s,a) \leq \frac{\log A}{\eta} + \eta \sum_{k=1}^{K}\sum_{a\in\mathcal{A}}\pi_k(a|s)|\hat{q}_k(s,a)|^2$$

$$\leq \frac{\log A}{\eta} + \eta K B_w^2.$$

We choose $\eta = \sqrt{\frac{\log A}{KB_w^2}}$, which is $\leq \frac{1}{B_w}$ whenever $K \geq \log A$, to get

$$\sum_{k=1}^{K}(\pi(a|s) - \pi_k(a|s))\hat{q}_k(s,a) \leq 2B_w\sqrt{K\log A}.$$

The proof is completed by dividing both sides by $K$, and taking an expectation with respect to $s \sim \mu^\pi$. $\qquad\square$

***Proof of Theorem 5.1.*** We work on the events of Lemmas 6.2 and Lemma 6.3 which, by a union bound, hold simultaneously with probability at least $1 - 2\delta$.

By Lemma 6.1 we have

$$\frac{1}{K}\sum_{j=1}^{K} J^\pi - \hat{J}_k = \frac{1}{K}\sum_{k=1}^{K}\mathbb{E}_{s\sim\mu^\pi}\left[(\pi(a|s) - \pi_k(a|s))\hat{q}_k(s,a)\right]$$

$$+ \frac{1}{K}\sum_{k=1}^{K}\mathbb{E}_{(s,a)\sim\mu^\pi\otimes\pi}\left[T^{\pi_k}\hat{q}_k(s,a) - \hat{J}_k - \hat{q}_k(s,a)\right].$$

By Lemma C.4, the first term on the right hand side bounded by $2B_w\sqrt{\frac{\log A}{K}}$. For the second term, by Lemma 6.3

$$\frac{1}{K}\sum_{k=1}^{K}\mathbb{E}_{(s,a)\sim\mu^\pi\otimes\pi}\left[T^{\pi_k}\hat{q}_k(s,a) - \hat{J}_k - \hat{q}_k(s,a)\right] \leq \frac{1}{K}\sum_{k=1}^{K}\left(2\beta\|\phi^{\mu^\pi}\|_{\hat{\Lambda}^{-1}} + \varepsilon_{B_w,B_{\boldsymbol{\theta}}}\right)$$

$$= 2\beta\|\phi^{\mu^\pi}\|_{\hat{\Lambda}^{-1}} + \varepsilon_{B_w,B_{\boldsymbol{\theta}}}.$$

We next observe that Lemma 6.2 implies

$$\frac{1}{K}\sum_{j=1}^{K} J^\pi - J^{\pi_k} \leq \frac{1}{K}\sum_{k=1}^{K} J^\pi - \hat{J}_k + \kappa_{B_w,B_{\boldsymbol{\theta}}}$$

$$\leq 2B_w\sqrt{\frac{\log A}{K}} + 2\beta\|\phi^{\mu^\pi}\|_{\hat{\Lambda}^{-1}} + \varepsilon_{B_w,B_{\boldsymbol{\theta}}} + \kappa_{B_w,B_{\boldsymbol{\theta}}}.$$

To complete the proof, we use the definition of $\beta$ and replace $\hat{\Lambda}^{-1}$ with $\frac{1}{N}\hat{\Lambda}_N^{-1}$. $\qquad\square$

# D  Proof of Lemma C.1

This section is dedicated to the proof of Lemma C.1. We start with the following general result whose purpose will be to show that policies in $\Pi$ are Lipchitz continuous in the parameter $\boldsymbol{\theta}$. This is similar to what is done in [Sherman et al., 2024].

**Lemma D.1.** *Let $\Theta \subset \mathbb{R}^d$ be a convex set and $\{f_\theta : \theta \in \Theta\}$ be a class of parameterized functions from $\mathbb{R}^d \to \mathbb{R}$. Suppose that the map $\theta \mapsto f_\theta(x)$ is continuously differentiable and that $\sup_{x\in\mathbb{R}^d, \|x\|_2\leq 1, \theta\in\Theta}\|\nabla_\theta f_\theta(x)\|_2 \leq L$ for some constant $L$. Then for any $\theta_1, \theta_2 \in \Theta$ and $s \in \mathcal{S}$,*

$$\|\pi_{\theta_1}(\cdot|s) - \pi_{\theta_2}(\cdot|s)\|_1 \leq 2L\|\theta_1 - \theta_2\|_2$$

*where*

$$\pi_\theta(a|s) = \frac{e^{f_\theta(\phi(s,a))}}{\sum_{a'} e^{f_\theta(\phi(s,a))}}$$

*and $\phi(s,a)$ is as in Assumption 3.3.*

*Proof.* Note that

$$\sup_{(s,a),\boldsymbol{\theta}\in\Theta} \|\nabla_{\boldsymbol{\theta}} f_{\boldsymbol{\theta}}(\phi(s,a))\|_2 \le L$$

since $\|\phi(s,a)\| \le 1$ under Assumption 3.3. Fix $\theta_1, \theta_2$ and a state $s \in \mathcal{S}$. Let $m = |\mathcal{A}|$ and let

$$J_\theta(s) = \begin{bmatrix} \nabla_\theta \pi_\theta(a_1|s) \\ \dots \\ \nabla_\theta \pi_\theta(a_m|s) \end{bmatrix} \in \mathbb{R}^{m \times d}$$

be the Jacobian of $\theta \mapsto \pi_\theta(\cdot|s)$. A straight forward computation shows that

$$\nabla_\theta \pi_\theta(a|s) = \pi_\theta(a|s) \left( \nabla_\theta f_\theta(\phi(s,a)) - \sum_{a'} \nabla_\theta f_\theta(\phi(s,a'))\pi_\theta(a'|s) \right). \tag{21}$$

Fix $t \in [0,1]$ and define $\theta_t = (1-t)\theta_1 + t\theta_2$. Let $J_{\theta_1,\theta_2}$ be the matrix $\int_0^1 J_{\theta_t} dt$. Then by the fundamental theorem of calculus

$$\|\pi_{\theta_1}(\cdot|s) - \pi_{\theta_2}(\cdot|s)\|_1 = \|J_{\theta_1,\theta_2}(\theta_1 - \theta_2)\|_1.$$

Now let $a$ be fixed. Using (21), we can bound the absolute value of the $a$-th entry of the vector $J_{\theta_1,\theta_2}(\theta_1 - \theta_2)$ as

$$
\begin{aligned}
|J_{\theta_1,\theta_2}(\theta_1 - \theta_2)_a| &\le \left| \int_0^1 \pi_{\theta_t}(a|s) \nabla_\theta f_{\theta_t}(\phi(s,a))^\top (\theta_1 - \theta_2) \right| \\
&\quad + \left| \int_0^1 \pi_{\theta_t}(a|s) \sum_{a'} \nabla_\theta f_{\theta_t}(\phi(s,a'))^\top (\theta_1 - \theta_2)\pi_{\theta_t}(a'|s) \right| \\
&\le \int_0^1 \pi_{\theta_t}(a|s)|\nabla_\theta f_{\theta_t}(\phi(s,a))^\top (\theta_1 - \theta_2)| dt \\
&\quad + \int_0^1 \pi_{\theta_t}(a|s) \sum_{a'} |\nabla_\theta f_{\theta_t}(\phi(s,a'))^\top (\theta_1 - \theta_2)|\pi_{\theta_t}(a'|s) dt \\
&\le L\|\theta_1 - \theta_2\|_2 \int_0^1 \pi_{\theta_t}(a|s) dt + L\|\theta_1 - \theta_2\|_2 \int_0^1 \pi_{\theta_t}(a|s) \sum_{a'} \pi_{\theta_t}(a'|s) dt \\
&= 2L\|\theta_1 - \theta_2\|_2 \int_0^1 \pi_{\theta_t}(a|s) dt
\end{aligned}
$$

where the second to last line we used Cauchy-Schwartz and $\|\nabla_\theta f_\theta(\phi(s,a))\| \le L$. Therefore

$$\|\pi_{\theta_1}(\cdot|s) - \pi_{\theta_2}(\cdot|s)\| = \sum_a |J_{\theta_1,\theta_2}(\theta_1 - \theta_2)_a| \le 2L\|\theta_1 - \theta_2\|_2 \int_0^1 \sum_a \pi_{\theta_t}(a|s) dt = 2L\|\theta_1 - \theta_2\|_2.$$

$\square$

Lemma D.1 can be applied to the restricted policy class $\Pi(B_{\boldsymbol{\theta}})$ by taking $\boldsymbol{\Theta}$ to be the $d$-dimensional Euclidean ball of radius $B_{\boldsymbol{\theta}}$ and the family $\{f_{\boldsymbol{\theta}} : \boldsymbol{\theta} \in \boldsymbol{\Theta}\}$ to be the set of functions $f_{\boldsymbol{\theta}}(\phi) = \phi^\top \boldsymbol{\theta}$ where $\|\phi\|_2 \le 1$. Clearly, we have $\|\nabla_{\boldsymbol{\theta}} f_{\boldsymbol{\theta}}(\phi)\|_2 \le 1$. Thus, Lemma D.1 shows that

$$\|\pi_{\boldsymbol{\theta}_1}(\cdot|s) - \pi_{\boldsymbol{\theta}_2}(\cdot|s)\|_1 \le 2\|\boldsymbol{\theta}_1 - \boldsymbol{\theta}_2\|_2 \tag{22}$$

for all $\pi_{\boldsymbol{\theta}_1}, \pi_{\boldsymbol{\theta}_2} \in \Pi(B_{\boldsymbol{\theta}})$.

With this we can now bound the $\varepsilon$-covering number of the function class $\mathcal{V}$.

**Lemma D.2.** *Let $\mathcal{V}$ be the function class* (16). *Then*

$$\log \mathcal{N}_\varepsilon(\mathcal{V}) \le d \log \left( 1 + 4B_w/\varepsilon \right) + d \log \left( 1 + 16B_w B_{\boldsymbol{\theta}}/\varepsilon \right).$$

*where $\mathcal{N}_\varepsilon(\mathcal{V})$ is the $\varepsilon$-covering number of $\mathcal{V}$ with respect to the norm $\|\cdot\|_\infty$*

*Proof.* First, consider two functions $q(\cdot,\cdot;\boldsymbol{w})$ and $q(\cdot,\cdot;\boldsymbol{w}')$ in $\mathcal{Q}(B_w)$. Using the normalization in Assumption 3.3

$$|q(s,a;\boldsymbol{w}) - q(s,a;\boldsymbol{w}')| = |\langle \phi(s,a), \boldsymbol{w} - \boldsymbol{w}' \rangle| \le \|\boldsymbol{w} - \boldsymbol{w}'\|_2$$

for any $(s, a)$. So for any fixed policy $\pi \in \Pi(B_{\boldsymbol{\theta}})$ we have

$$
\begin{aligned}
|v(s; \pi, \boldsymbol{w}) - v(s; \pi, \boldsymbol{w}')| &\leq \max_s \left| \sum_a \pi(a|s)(q(s, a, \boldsymbol{w}) - q(s, a, \boldsymbol{w}')) \right| \\
&\leq \max_{s,a} |q(s, a, \boldsymbol{w}) - q(s, a, \boldsymbol{w}')| \\
&\leq \|\boldsymbol{w} - \boldsymbol{w}'\|_2.
\end{aligned}
$$

On the other hand, for a fixed $\boldsymbol{w}$, and separate policies $\pi_{\boldsymbol{\theta}}, \pi_{\boldsymbol{\theta}'} \in \Pi$,

$$
\begin{aligned}
|v(s, \pi, \boldsymbol{w}) - v(s, \pi', \boldsymbol{w})| &\leq \max_s \left| \sum_a (\pi_{\boldsymbol{\theta}}(a|s) - \pi_{\boldsymbol{\theta}'}(a|s)) q(s, a; \boldsymbol{w}) \right| \\
&\leq B_w \max_s \|\pi_{\boldsymbol{\theta}}(\cdot|s) - \pi_{\boldsymbol{\theta}'}(\cdot|s)\|_1 \\
&\leq 4 B_w \|\boldsymbol{\theta} - \boldsymbol{\theta}'\|_2
\end{aligned}
$$

where the second line used $|q(s, a); \boldsymbol{w})| \leq B_w$ and the last line used (22). Thus it holds for any $v(\cdot, \pi, \boldsymbol{w})$, $v(\cdot, \pi', \boldsymbol{w}') \in \Pi$,

$$
|v(s, \pi, \boldsymbol{w}) - v(s, \pi', \boldsymbol{w}')| \leq |v(s, \pi, \boldsymbol{w}) - v(s, \pi, \boldsymbol{w}')| + |v(s, \pi, \boldsymbol{w}') - v(s, \pi', \boldsymbol{w}')| \tag{23}
$$
$$
\|\boldsymbol{w} - \boldsymbol{w}'\|_2 + 4 B_w \|\boldsymbol{\theta} - \boldsymbol{\theta}'\|_2.
$$

Now, using a standard result concerning the covering number of the $d$-dimensional Euclidian ball (Lemma E.2), we can construct an $\frac{\varepsilon}{2}$ covering of the euclidean ball $d$-dimensional euclidean ball of radius $B_w$ with cardinality at most $(1 + 4 B_w / \varepsilon)^d$ and an $\frac{\varepsilon}{8 B_w}$ covering of the Euclidean ball of radius $B_{\boldsymbol{\theta}}$ with cardinality not exceeding $(1 + 16 B_w B_{\boldsymbol{\theta}} / \varepsilon)^d$. Let $\mathcal{V}_\varepsilon$ be the members of $\mathcal{V}$ parameterized by members $(\boldsymbol{w}', \boldsymbol{\theta}')$ of the Cartesian product of these two coverings. Then

$$
\log \mathcal{N}_\varepsilon(\mathcal{V}) = \log |\mathcal{V}_\varepsilon| \leq d \log(1 + 4 B_w / \varepsilon) + d \log(1 + 16 B_w B_{\boldsymbol{\theta}} / \varepsilon),
$$

and by (23), for any $v(\cdot; \pi_{\boldsymbol{\theta}}, \boldsymbol{w}) \in \mathcal{V}$ we can find $v(\cdot; \pi_{\boldsymbol{\theta}'}, \boldsymbol{w}') \in \mathcal{V}_\varepsilon$ with

$$
\begin{aligned}
|v(s; \pi_{\boldsymbol{\theta}}, \boldsymbol{w}) - v(s, \pi_{\boldsymbol{\theta}'}, \boldsymbol{w}')| &\leq \|\boldsymbol{w} - \boldsymbol{w}'\|_2 + 4 B_w \|\boldsymbol{\theta} - \boldsymbol{\theta}'\|_2 \\
&\leq \frac{\varepsilon}{2} + 4 B_w \cdot \frac{\varepsilon}{8 B_w} = \varepsilon.
\end{aligned}
$$

$\square$

***Proof of Lemma C.1.*** Fix $\{v_k\}_{k=1}^K \subset \mathcal{V}(B_w, B_{\boldsymbol{\theta}})$ Appealing to the uniform concentration of self-normalized processes (Lemma E.5) and using that $\|v\|_\infty \leq B_w$ for any $v \in \mathcal{V}$ we have, for fixed $k$,

$$
\left\| \sum_{i=1}^N \phi(s_i, a_i) \left( P_{s_i, a_i} v_k - v_k(s_i') \right) \right\|_{\hat{\Lambda}^{-1}}^2 \leq 4 B_w^2 \left( \frac{d}{2} \log \left( \frac{K(N+1)}{\delta} \right) + \log \mathcal{N}_\varepsilon(\mathcal{V}) \right) + 8 N^2 \varepsilon^2
$$

holds with probability $1 - \frac{\delta}{K}$. Now, substituting the bound for $\log \mathcal{N}_\varepsilon(\mathcal{V})$ from Lemma D.2, the upper bound becomes

$$
4 B_w^2 \left( \frac{d}{2} \log \left( \frac{K(N+1)}{\delta} \right) + d \log(1 + 4 B_w / \varepsilon) + d \log(1 + 16 B_w B_{\boldsymbol{\theta}} / \varepsilon) \right) + 8 N^2 \varepsilon^2.
$$

Taking $\varepsilon = 1/N$ followed by a union bound for over $k = 1, \dots, K$ we complete the proof. $\square$

# E  Auxiliary Lemmas

**Lemma E.1.** *Let $\{X_k\}_{k \geq 1}$ be a sequence of vectors in $\mathbb{R}^A$. Set $\pi_1(a) = \frac{1}{A}$ for all $a \in \mathcal{A}$ and for each $k \geq 2$,*

$$
\pi_{k+1}(a) = \frac{\pi_k(a) \exp(\eta X_k(a))}{\sum_{a' \in \mathcal{A}} \pi_k(a') \exp(\eta X_k(a'))}
$$

*for some positive stepsize $\eta$ satisfying $\eta X_k(a) \geq -1$ for all $k$ and $a \in \mathcal{A}$. Then for any fixed $\pi^* \in \Delta(\mathcal{A})$,*

$$
\sum_{k=1}^K \langle \pi^* - \pi_k, X_k \rangle \leq \frac{\log A}{\eta} + \eta \sum_{k=1}^K \sum_{a \in \mathcal{A}} \pi_k(a) X_k(a)^2.
$$

*Proof.* Define $Z_k = \sum_{a' \in \mathcal{A}} \pi_k(a') \exp(\eta X_k(a'))$. Then using the inequality $e^x \leq 1 + x + x^2$ which holds for all $x \leq 1$ followed by $\log(1+x) \leq x$ we have

$$\log Z_k = \log \left( \sum_{a' \in \mathcal{A}} \pi_k(a') \exp(\eta X_k(a')) \right) \leq \log \left( 1 + \sum_{a \in \mathcal{A}} \pi_k(a) \eta X_k(a) + \sum_{a \in \mathcal{A}} \pi_k(a) \eta^2 X_k(a)^2 \right)$$

$$\leq \eta \sum_{a \in \mathcal{A}} \pi_k(a) X_k(a) + \eta^2 \sum_{a \in \mathcal{A}} \pi_k(a) X_k(a)^2.$$

Thus,

$$D_{KL}(\pi^* || \pi_{k+1}) - D_{KL}(\pi^* || \pi_k) = \sum_{a \in \mathcal{A}} \pi^*(a) \log \left( \frac{\pi^*(a)}{\pi_{k+1}(a)} \right) - \sum_{a \in \mathcal{A}} \pi^*(a) \log \left( \frac{\pi^*(a)}{\pi_k(a)} \right)$$

$$= \sum_{a \in \mathcal{A}} \pi^*(a) \log \left( \frac{\pi^*(a) Z_k \exp(-\eta X_k(a))}{\pi_k(a)} \right) - \sum_{a \in \mathcal{A}} \pi^*(a) \log \left( \frac{\pi^*(a)}{\pi_k(a)} \right)$$

$$= \log Z_k - \eta \sum_{a \in \mathcal{A}} \pi^*(a) X_k(a)$$

$$\leq \eta \sum_{a \in \mathcal{A}} \pi_k(a) X_k(a) + \eta^2 \sum_{a \in \mathcal{A}} \pi_k(a) X_k(a)^2 - \eta \sum_{a \in \mathcal{A}} \pi^*(a) X_k(a).$$

Rearranging and summing from $k = 1$ to $K$,

$$\eta \sum_{k=1}^{K} \sum_{a \in \mathcal{A}} (\pi^*(a) - \pi_k(a)) X_k(a) \leq \sum_{k=1}^{K} D_{KL}(\pi^* || \pi_k) - D_{KL}(\pi^* || \pi_{k+1}) + \eta^2 \sum_{k=1}^{K} \sum_{a \in \mathcal{A}} \pi_k(a) X_k(a)^2$$

$$\leq D_{KL}(\pi^* || \pi_1) + \eta^2 \sum_{k=1}^{K} \sum_{a \in \mathcal{A}} \pi_k(a) X_k(a)^2.$$

Since $\pi_1(a) = \frac{1}{A}$ for all $a$,

$$D_{KL}(\pi^* || \pi_1) = \sum_{a \in \mathcal{A}} \pi^*(a) \log(A \pi^*(a)) \leq \log A.$$

Plugging this bound in above and dividing both sides by $\eta$ we complete the proof. $\square$

**Lemma E.2** (Covering Number of Euclidean Ball). *For any $\varepsilon > 0$, the $\varepsilon$-covering number of the Euclidean ball in $\mathbb{R}^d$ with radius $R > 0$ is upper bounded by $(1 + 2R/\varepsilon)^d$*

*Proof.* See [Jin et al., 2020] Lemma D.5. $\square$

**Lemma E.3** (Projection Bound). *Let $\{\phi_i\}_{i \geq 1}$ be a sequence in $\mathbb{R}^d$ with $\|\phi_i\|_2 \leq 1$ and $\{a_i\}_{i \geq 1}$ be a sequence of real numbers with $|a_i| \leq A$. Define*

$$\Lambda_n = \sum_{i=1}^{n} \phi_i \phi_i^\top + I.$$

*We have*

$$\left\| \sum_{i=1}^{n} a_i \phi_i \right\|_{\Lambda_n^{-1}} \leq A\sqrt{n}.$$

*Proof.* See [Zanette et al., 2020] Lemma 8. $\square$

**Lemma E.4** (Concentration of Self-Normalized Processes). *Let $(X_t)_t$ be a real-valued martingale difference sequence adapted to filtration $(\mathcal{F}_t)_t$. Suppose that $X_t$ is $\sigma$-subgaussian conditioned on $\mathcal{F}_{t-1}$ i.e.,*

$$\log \mathbb{E}[e^{\lambda X_t} | \mathcal{F}_{t-1}] \leq \frac{\lambda^2 \sigma^2}{2}.$$

*Let $(\phi_t)_t$ be an $\mathbb{R}^d$-valued predictable process. Assume $\Lambda_0 \in \mathbb{R}^{d \times d}$ is positive definite and let $\Lambda_t = \Lambda_0 + \sum_{s=1}^{t} \phi_s \phi_s^\top$. Then for any $\delta > 0$, with probability at least $1 - \delta$ we have*

$$\left\| \sum_{s=1}^{t} \phi_s X_s \right\|_{\Lambda_t^{-1}}^2 \leq 2\sigma^2 \log \left( \frac{\sqrt{det(\Lambda_t) det(\Lambda_0)}}{\delta} \right).$$

*Proof.* See [Jin et al., 2020] Lemma D.3.  □

**Lemma E.5** (Lemma D.4 in [Jin et al., 2020])**.** *Let $\{x_t\}_{t=1}^{\infty}$ be a stochastic process on state space $\mathcal{S}$ with corresponding filtration $\{\mathcal{F}_t\}_{t=1}^{\infty}$. Let $\{\phi_t\}_{t=1}^{\infty}$ be an $\mathbb{R}^d$-valued stochastic process where $\phi_t \in \mathcal{F}_{t-1}$ and $\|\phi_t\| \leq 1$. Let $\Lambda_k = \sum_{t=1}^{k} \phi_t \phi_t^\top$. Then for any $\delta > 0$, with probability at least $1 - \delta$, for all $k \geq 0$, and any $V \in \mathcal{V}$ so that $\|V\|_\infty \leq H$, we have:*

$$\left\| \sum_{t=1}^{k} \phi_t \left( V(x_t) - \mathbb{E}[V(x_t)|\mathcal{F}_{t-1}] \right) \right\|_{\Lambda_k^{-1}}^2 \leq 4H^2 \left( \frac{d}{2} \log \left( \frac{k+\lambda}{\lambda} \right) + \log \frac{\mathcal{N}_\varepsilon(\mathcal{V})}{\delta} \right) + \frac{8k^2\varepsilon^2}{\lambda},$$

*where $\mathcal{N}_\varepsilon(\mathcal{V})$ is the $\varepsilon$-covering number of $\mathcal{V}$ with respected to the distance $dist(V, V') = \sup_x |V(x) - V'(x)|$.*

