# OpenReview forum: "Offline Actor-Critic for Average Reward MDPs"
_NeurIPS.cc/2025/Conference — NeurIPS 2025 poster_

### Official Review · Reviewer_ni8W · 2025-06-17

**Clarity:** 4
**Significance:** 4
**Originality:** 4
**Rating:** 5
**Confidence:** 4

**Summary:**

This paper proposes a pessimistic actor-critic algorithm for offline reinforcement learning (RL) in infinite-horizon average reward Markov Decision Processes (AMDPs), using linear function approximation of the value function. The key technical contribution is a novel, computationally efficient critic step that avoids solving successive sequence of regression problems by directly solving a fixed-point Bellman equation using a linear optimization formulation with convex quadratic constraints. The algorithm is proven to be sample-efficient, with a near-optimal sample complexity of $\mathcal{\tilde{O}}(\epsilon^{-2})$, under a weak notion of data coverage (the feature coverage ratio). This is the first such result in the offline average-reward setting that matches the best known rate and only requires for the MDP induced by all policies to be unichain.

**Questions:**

N/A

**Ethical Concerns:**

["NO or VERY MINOR ethics concerns only"]

**Final Justification:**

Accept because an important problem (offline RL in infinite-horizon, average reward setting) considered, clever algorithm, significant improvement in sample complexity from previous SOTA, well-written

Not Strong Accept because the problem considered is a logical next step in the offline RL line of work and the proof techniques presented are significant but not ground breaking

**Limitations:**

Yes, the authors have adequately addressed the limitations and potential negative societal impact of their work.

**Quality:**

4

**Strengths And Weaknesses:**

**Strengths:**
- The paper addresses the problem of offline RL infinite-horizon, average reward setting, an important gap in literature
- The analysis requires mild assumption compared to prior literature - it requires only unichain (instead of uniformly ergodic) and a relatively weak notion of the feature coverage ratio
- The algorithm cleverly constructs a pessimistic estimate of the average reward by solving a single fixed-point Bellman equation and address computational efficiency through the reformulation as a linear optimization with convex quadratic constraints
- Significantly improves upon the sample complexity in comparison to prior offline RL literature and matches the best known in online RL domain
- Well written paper with a clear presentation of the proof sketch and a through discussion of the assumptions used

**Weaknesses:**
- A discussion on the effects of scaling state and action space might be nice to include
- Insight into how the current work may be extended to the general non-linear function approximation setting would make the paper comprehensive
- While weaker than uniform ergodicity, the unichain assumption may not always hold. As mentioned in the section on future directions, it would be interesting to see how this affects algorithm design and analysis.

---

> ### Author Rebuttal · Authors · 2025-07-31
>
> Thank you for your time and detailed review. Please see our responses below.
>
> > A discussion on the effects of scaling state and action space might be nice to include
>
> The bound in Theorem 5.1 only depends on the size of the action space, and only scales logarithmically with this number. Importantly, the bound does not depend directly on the size of the state space at all, only on the dimension of the feature map $d$. We will be sure to add this discussion in the revision.
>
> > Insight into how the current work may be extended to the general non-linear function approximation setting would make the paper comprehensive
>
> The main obstacle in generalizing our algorithm in its exact form to more general function approximation is the confidence ellipsoid, $\lVert \xi \rVert_{\hat{\Lambda}} \leq \beta$ . This is based on martingale concentration techniques from the linear MDP literature (Lemma B.1). These techniques allow us to very precisely quantify the level of uncertainty in the dataset. It is not yet clear how to do this for more general function approximation like neural networks.
>
> However a promising avenue may be to adapt techniques from papers such as [Xie et al., 2021] and [Cheng et al., 2022]. Specifically, one can consider removing the pessimism parameter $\xi$ and considering a Lagrangian relaxation of the critic's problem. Specifically, in episode $k$ we may ask the critic to find a function $q_k\in \mathcal{Q}$ minimizing $$J + \frac{\lambda}{N} \sum_{i = 1}^N (q(s_i, a_i) + J - r(s_i, a_i) - E_{a \sim \pi_k}[q(s_i', a)])^2,$$ where $\mathcal{Q}$ is some suitable class for function approximation, $\lambda$ is a Lagrange multiplier which can be tuned to control the degree of pessimism, and $$\frac{1}{N} \sum_{i = 1}^N (q(s_i, a_i) + J - r(s_i, a_i) - E_{a \sim \pi_k}[q(s_i', a)])^2$$ is the average square TD error. The main challenge we see here, which is not present in the discounted setting, is the presence of the additional parameter $J$ meant to estimate the average reward of policy $\pi_k$. Ideally, $J$ should depend on $q$ in such a way such that $J(q_k)$ is a pessimistic estimate of the average reward of policy $\pi_k$. We leave this interesting problem for future work.
>
>
> >While weaker than uniform ergodicity, the unichain assumption may not always hold. As mentioned in the section on future directions, it would be interesting to see how this affects algorithm design and analysis.
>
> There are two main difficulties we come across without the unichain assumption. The first is that, in this case, $J^{\pi}$ not be constant and could depend the initial state. The formulation of the optimization problem (6) may need to change to reflect this. The second is that, without Assumption 3.1, the true Q-functions may be unbounded. The stability of our algorithm relies on the members of the function class $\mathcal{Q}(B_w)$ being bounded. If the true Q functions are unbounded, but the elements of $\mathcal{Q}(B_w)$ are, it may be unreasonable to expect the completeness and approximate realizability assumptions to hold. We will add the discussion in the revision.

---

### Official Review · Reviewer_vssp · 2025-06-29

**Clarity:** 3
**Significance:** 3
**Originality:** 3
**Rating:** 4
**Confidence:** 4

**Summary:**

The authors proposed an actor-critic offline reinforcement learning algorithm for average reward MDP, which first achieved O(\eps^{-2}) sample complexity. As a paper focusing on the theoretical guarantee, the authors provided detailed proof and discussion on the relationship with the previous paper.

**Questions:**

Please refer to the clarity and Originality Parts for the questions. This paper is technically solid. I would increase the score if the author could confirm with me the question mentioned above.

**Ethical Concerns:**

["NO or VERY MINOR ethics concerns only"]

**Final Justification:**

Reason to borderline accept: close the gap of RL theory in the offline setting, which is important to the community.
Reason not to accept or strong accept: Although the proposed algorithms achieved faster convergece rate, the proof technique is somehow not groundbreaking

**Limitations:**

yes

**Quality:**

3

**Strengths And Weaknesses:**

Quality: 3
The main paper quality is good. The author provided an actor-critic framework to the offline RL problem. Decompose the sub-optimality into actor and critic parts. For the actor parts, the author adopts the Natural policy gradient method and achieves an optimization error convergence with order O(1/\sqrt(k)). For the critic part, the authors applied Bellman equation regression with pessimism parameters. The proof sketch and details are solid. Moreover, the author discusses the assumption and previous work in depth, which is helpful.

Clarity: 2
Despite the algorithm and proof being solid in this paper, some errors or typos may lead to misunderstanding.

(1)	In the section 4.2 policy update, it is better to make it clear that the update follows Mirror Ascent with KL divergence or Natural policy gradient instead of general gradient ascent to make a better understanding
(2)	In the proof of Lemma B.3, the notation \bar{\omega}_k and \omega_k^* are mixed up? Make it hard to read the proof of Lemma B.3
(3)	In the proof of Theorem 5.1, line 610, the optimization error term is missing.

Significance: 3
The authors discussed the convergence rate of the offline RL under the Average MDP setting, which is an important topic in RL theory and closes the gap.

Originality: 2
Although the discussed problem is important to the community. However, it seems the novelty is somehow limited. The actor-critic framework is followed from [Zanette 2021]. Although I agree that [Zanette 2021] discussed the episodic setting and the induction technique is invalid in the average setting. However, it seems that the idea of proof procedure (Lemma 6.2 and 6.3) is similar to the FOPO algorithms' proof in [Wei et al, 2021]. The only difference is that [Wei et al, 2021] discusses online problems with Optimism, and this paper discusses offline problems with pessimism.

It would be great to highlight the contribution of the technical part of this paper if the authors could answer the following question.

The author mentioned that [Gabbianelli et al., 2024] is the only paper that also discusses offline RL under average MDP. However, they only achieve O(\eps^{-4}). Could the author explain what the new idea or technique is adopted to improve the result to O(\eps^{-2})?

---

> ### Author Rebuttal · Authors · 2025-07-31
>
> Thank you very much for your time and detailed review. Please see our responses to your questions below.
>
> ### Clarity
> There are a few typos in the proof. Thank you for pointing these out!
>
> The $w_k^*$ appearing in the proof of Lemma B.3 is a typo. It should be $\bar{w}_k$ throughout the proof. We have corrected this.
>
> The second inequality in line 610 should read $$\frac{1}{K} \sum_{k = 1}^{K} J^{\pi} - \hat{J_k} + \kappa_{B_w, B_{\theta}} \leq 2B_w \sqrt{\frac{\log A}{K}} + 2\beta \lVert\phi^{\mu^{\pi}}\rVert_{\hat{\Lambda}^{-1}} + \varepsilon_{B_w, B_{\theta}} + \kappa_{B_w, B_{\theta}}.$$
> We have corrected this as well.
>
> We will also update section 4.2 to be clear that the policy update follows mirror ascent with KL divergence regularization.
>
> ### Originality
>
> > However, it seems that the idea of proof procedure (Lemma 6.2 and 6.3) is similar to the FOPO algorithms' proof in [Wei et al, 2021]. The only difference is that [Wei et al, 2021] discusses online problems with Optimism, and this paper discusses offline problems with pessimism.
>
> Although our analysis is partly inspired by [Wei et al., 2021], we would like to point out some notable differences below.
>
>
> 1. Unlike [Wei et al, 2021], we do not require the perfect linear MDP assumption. Specifically, we only assume the approximate realizability and restricted closedness properties. Additional work is required for the analysis to handle this level of generality. For instance, in the proof of Lemma 6.2, we cannot assume that $w_k^{\star}$ parameterizes the true Q-function $q^{\pi_k}$ exactly. So the function $q_k^{\star}(s, a) = \phi(s, a)^{\top}w_k^{\star}$ need not be a solution to the Bellman equation. This results in the additional term $\hat{\Lambda}^{-1} \sum_{i = 1}^{N} \phi(s_i, a_i) \Delta_{k}(s_i,a_i)$ in equation (17), where $\Delta_k(s, a) = q_k^{\star}(s, a) + J_k^{\star} - T^{\pi_k}q_k^{\star}(s, a)$ is the irreducible Bellman error due to realizability holding only approximately. When arguing that $w_k^{\star}, J_k^{\star}$ is feasible for the optimization problem (6) with high probability, this term needs to be accounted for. This is why the additional factor of $\kappa_{\mathcal{Q}(B_w), \Pi} \sqrt{N}$ appears in the definition of the confidence parameter $\beta$. A similar error term also appears in Lemma 6.3 due to the restricted closedness assumption.
>
> 2. Because our algorithm is policy based, our analysis is built on the extended performance difference lemma (Lemma 6.1). This differs from the proof structure in [Wei et al., 2021] since their algorithm takes greedy actions. More specifically, the way we decompose the sub-optimality in step 3 of the proof sketch is quite different. We decompose the sub-optimality into an optimization term and a bias term. Our Lemma 6.3 focuses on controlling the bias in the estimate of the Q-function for each policy $\pi_k$ encountered during the algorithm's execution. Such a statement does not appear in [Wei et al., 2021] because thier algorithm is designed to approximate the solution to the Bellman optimality equation in each episode, rather than the solution for a fixed policy. They then need to argue that the bias in their estimate of the *optimal* Q function is not too large. Therefore, their regret analysis is based entirely on the Bellman optimality equation and martingale difference concentration rather than a performance difference lemma.
>
> >The author mentioned that [Gabbianelli et al., 2024] is the only paper that also discusses offline RL under average MDP. However, they only achieve O(\eps^{-4}). Could the author explain what the new idea or technique is adopted to improve the result to O(\eps^{-2})?
>
> Thank you for the question. To explain our new technique, we first briefly summarize the main ideas of [Gabbianelli et al., 2024] using their notation. In their work, they assume a linear MDP structure, realized by $\Psi \in R^{d \times |S|}$, $w \in R^d$ satisfying $$r(s, a) = \langle \phi(s, a), w \rangle, \qquad P(s'|s, a) = \langle \phi(s, a), \psi(s') \rangle,$$ where $\psi(s')$ is the $s'$ column of $\Psi$. To adapt to average reward case, they also assume there is some $\varrho \in R^d$ with $\langle \phi(s, a), \varrho\rangle = 1$. Assuming that state action pairs are drawn i.i.d from some distribution $\mu_{B}$, the population covariance is $\Lambda = E_{\mu_B}[\phi(s, a)\phi(s, a)^{\top}]$. Under their linear MDP assumption, they observe the average reward (they call $\rho^{\pi}$) can be written as $\rho^{\pi} = \langle \lambda^{\pi}, w \rangle$ where $\lambda^{\pi} = E_{(s, a)\sim \mu^{\pi} \otimes \pi}[\phi(s, a)]$. They then formulate the problem as a linear program with Lagrangian $$f(\beta, \pi; \rho, \theta) = \rho + \langle \beta, \Lambda[w + \Psi v_{\theta, \pi} - \theta - \rho \varrho]\rangle.$$
> Here $v_{\theta, \pi}(s) = \sum_{a}\pi(a|s)\langle \phi(s, a), \theta \rangle$ and $\beta = \Lambda^{-1}\lambda$ is a reparameterization used so that knowledge of $\Lambda$ is not necessary to run the algorithm. Their main idea is to solve the saddle point problem $$\min_{\rho, \theta}\max_{\beta, \pi} f(\beta, \pi; \rho, \theta).$$ Note, importantly, that this is a population level problem. Since they only assume access to i.i.d. samples from $\mu_{B}$, they solve the problem via stochastic gradient descent-ascent.
>
>  The main bottleneck in the analysis of [Gabbianelli et al., 2024] is their algorithm's double-loop structure that, in each outer loop $t$, uses an inner-loop to solve $$\min_{\rho, \theta} f(\beta_t, \pi_t; \rho, \theta)$$ nearly exactly using $O(\varepsilon^{-2})$ samples.Then as they need $O(\varepsilon^{-2})$ outer-loop total iterations, they could only get the $O(\varepsilon^{-4})$ upper bound.
>
> The key reason for the sample complexity improvement in our paper is the efficient use of the entire dataset for the critic's optimization problem (6). Namely, the data to construct this problem only needs to be sampled once and is then re-used in each iteration of our algorithm. This avoids the inner-loop that uses additional samples.  The only issue that needs to be dealt with is the statistical correlation between iterates of the algorithm arising from data re-use. This correlation is handled in the analysis using the $\epsilon$-net covering technique.
>
> We would also like to point out that the linear program structure of [Gabianelli et al., 2024] crucially relies on the linear MDP structure. Our use of the constrained optimization approach is the key reason we can relax the assumptions to cover restricted closedness and approximate realizability.

---

> ### Comment · Reviewer_vssp · 2025-08-06
>
> The authors have clearly answered my question (1) the difference that I didn't notice compared with [Wei et al, 2021] (2) The reason why the proposed algorithm achieved much faster convergence rate than [Gabbianelli et al., 2024].
>
> I highly encourage the authors to explain the advantage over [Gabbianelli et al., 2024] in their revised version.
> I will keep my positive score.

---

### Official Review · Reviewer_oazj · 2025-07-02

**Clarity:** 4
**Significance:** 4
**Originality:** 3
**Rating:** 5
**Confidence:** 3

**Summary:**

The authors propose an actor-critic method for efficient offline RL in average reward MDPs with an intractably large or infinite number of states. They consider the linear function approximation framework, and make assumptions on the expressivity of their function approximator including approximate $q^{\pi}$-realizability and Bellman completeness under any policy. Inspired by the actor-critic algorithm of Zanette et. al [2021] for the finite-horizon setting, for steps $k=1,...,K$ their algorithm obtains the critic’s weight $w_k$ as the solution of an optimization problem, then updates the actor parameter $\theta_k$ such that the resulting actor policy in any state is equivalent to one step of mirror-ascent with KL regularization and critic evaluation $\hat{q}_k(s,\cdot)=\langle \phi(s,\cdot),w_k\rangle$ as the gradient. Similar to Zanette et al [2021], their optimization problem is constructed in each step using samples from the offline dataset, and constrains the critic weights to be equivalent to the perturbed solution of a ridge regression problem – with the sample estimate of $q^{\pi_k}$ as the target. Furthermore, in the absence of the backward induction property enjoyed by finite-horizon MDPs, and the presence of the cumulative return $J^{\pi}$ in the Bellman expression of $Q$-functions in the average-reward MDP setting, they propose two modifications to the optimization problem: 1) a new decision variable $J$ to estimate $J^{\pi_k}$ and 2) a direct parametrization of an estimator for the state-value function $v^{\pi_k}$ based on the critic’s weights. With these modifications, they prove $O(\varepsilon^{-2})$ sample complexity guarantee under the weakest known data collection and coverage conditions. Precisely, they do not require that the dataset is generated i.i.d or by a single behaviour policy, and they only require the dataset to properly cover a single direction in the feature space. Their sample complexity guarantee also scales linearly (rather than quadratically) with the coverage parameter.

**Questions:**

I have just one question:
* Do the authors assume that the initial state is fixed?

Some typos:
1. L286: “However, their algorithm’s...” should be “However, their algorithm...”
2. L299: “...while we study the more general function...” should be “...while we study the more general linear function...”
3. After L335, the second inequality: $(s,a)\sim\mu^{\pi}$ should be $s \sim\mu^{\pi}$

**Ethical Concerns:**

["NO or VERY MINOR ethics concerns only"]

**Final Justification:**

I maintain my positive score of the paper due to the strengths listed above, and a satisfactory discussion with the authors. The paper addresses an important problem in offline RL, with adequate theoretical justifications and very few typos. Furthermore, the authors acknowledged my comments on, and agreed to clarify potentially misleading claims about the computational efficiency of their method.

**Quality:**

3

**Strengths And Weaknesses:**

As is evident in the long summary, I did enjoy reading the paper. The problem of offline RL in infinite-horizon MDPs with intractably many states, under weaker conditions like $q^{\pi}$-realizability and Bellman completeness, is indeed a relevant open problem in the offline RL literature. The paper addresses this problem with an $O(\varepsilon^{-2})$ sample complexity guarantee and their bound scales linearly with the single-direction coverage parameter, rather than quadratically – which is more common in related works.

There is no notable weakness. However, I would like to highlight that the algorithm is not fully computationally efficient. It is true that Algorithm 1 can be efficiently implemented with the help of an interior-point method to solve the optimization problem in step 5. While interior-point methods are highly precise, they are not exact. That said, the theoretical guarantees provided in the paper hold under the assumption that there exists an optimization solution which provides an exact solution to step 5. I strongly recommend that the authors add a comment to clarify this, especially in the abstract and other parts of the main text with explicit claims about computational efficiency.

---

> ### Author Rebuttal · Authors · 2025-07-31
>
> We thank the reviewer for their time and positive review. Please see our response below.
>
> >However, I would like to highlight that the algorithm is not fully computationally efficient. It is true that Algorithm 1 can be efficiently implemented with the help of an interior-point method to solve the optimization problem in step 5. While interior-point methods are highly precise, they are not exact. That said, the theoretical guarantees provided in the paper hold under the assumption that there exists an optimization solution which provides an exact solution to step 5. I strongly recommend that the authors add a comment to clarify this, especially in the abstract and other parts of the main text with explicit claims about computational efficiency.
>
> Thank you for raising this point. While we assumed the use of exact solutions, inexact solutions still only incur an additive error proportional to the accuracy of the solution. Specifically, suppose that the optimization problem (6) is solved to within accuracy $\delta$, so that $J_k$ is within $\delta$ of the optimal solution and the constraint violation is also at most $\delta$. The computational complexity of many existing algorithms to obtain such a solution solution scale as $\log\delta^{-1}$ [Nesterov and Nemirovskii, 1994].  Then additional error will be incurred in Lemma 6.2 where we will instead have
> $$J_k \leq J_k^* + \delta \le J^{\pi_k} + \kappa_{\mathcal{Q}(B_w), \Pi} + \delta,\quad \forall k \in [K].$$ There will also be an error incurred in Lemma 6.3 due to the constraint violations, which can be updated to read
> $$|E_{(s, a) \sim \mu} [q_k(s, a) - \phi(s, a)^{\top}\bar{w_k}]| \leq   2(\beta + \delta) \lVert\phi^{\mu}\rVert_{\hat{\Lambda}^{-1}} + \delta.$$
> Here the factor of $2\delta$ that is a coefficient of $\lVert\phi^{\mu}\rVert_{\hat{\Lambda}^{-1}}$ is due to the violation of the quadratic constraints and the additive factor is from a violation of the linear constraint. We will add a clarifying statement in the main text accordingly.
>
> >Do the authors assume that the initial state is fixed?
>
> We do not need to assume that the initial state is fixed. It may be random. Under assumption 3.1, the average reward $J^{\pi}$ does not depend on the initial state. Therefore, the identity of the initial state, or distribution if it is random, does not change the problem or our results.
>
> Thank you for pointing out the typos! We will correct these in the revision.
>
> [Yurii Nestorov and Arkadii Nemirovskii, 'Interior-Point Polynomial Algorithms in Convex Programming', Society for Industrial and Applied Mathematics, 1994]

---

> > ### Comment · Reviewer_oazj · 2025-08-04
> >
> > **Re Discussion on Computational Efficiency of Alg 1**: Thanks for the concise response. However, my comment is regarding the author's remark "To our knowledge, this is the first result with optimal εdependence in the offline average reward setting via a computationally efficient algorithm." in the abstract, and similar remarks on Algorithm 1 being "fully computationally efficient" in the main text. While the authors are free to discuss error propagation of an inexact (and fully computationally efficient) version of Algorithm 1, the above remarks are presently misleading due to my earlier comments.

---

> > > ### Author Response · Authors · 2025-08-05
> > >
> > > Thank you for the clarification. We will remove the phrase "via a computationally efficient algorithm" from the last sentence of the abstract. We will also modify the main text to be clear that an implementation of the algorithm with exact solutions to the critic’s optimization problem is not fully computationally efficient. We will add an additional discussion about error propagation for an inexact, computationally efficient version in the revision.

---

### Official Review · Reviewer_wWmr · 2025-07-03

**Clarity:** 3
**Significance:** 3
**Originality:** 3
**Rating:** 4
**Confidence:** 3

**Summary:**

The paper proposes a pessimistic actor–critic algorithm for offline reinforcement learning in infinite-horizon average-reward MDPs (AMDPs) with linear function approximation.
- The critic obtains a lower-bound (pessimistic) estimate of the average reward and its policy gradient by solving a single linear program with convex quadratic constraints that represents the fixed-point Bellman equation, avoiding the step-by-step regressions common in episodic settings .
- The actor performs a conservative policy-improvement step on a soft-max policy class. Repeating the alternation $K$ times and outputting a mixture of intermediate policies yields an $\varepsilon$-optimal policy (up to model-mis-specification error) with sample complexity $O(\varepsilon^{-2})$—the first optimal-rate result for offline AMDPs .
- The theory relies on (i) unichain dynamics with bounded Q-span, and (ii) approximate realizability + Bellman restricted closedness of the linear feature class .
- Compared with the only prior provably efficient offline AMDP method, PDOR, the new algorithm improves the ε-dependence from $O(\varepsilon^{-4})$ to $O(\varepsilon^{-2})$ and removes the need for exact linear MDP structure .

**Questions:**

The conclusion suggests extending to “general function approximation.” What obstacles prevent directly replacing the linear critic with a neural network trained by convex-constrained regression?

**Ethical Concerns:**

["NO or VERY MINOR ethics concerns only"]

**Limitations:**

yes

**Quality:**

3

**Strengths And Weaknesses:**

# Strengths
- Rigorous proofs culminating in Theorem 5.1 with explicit finite-sample bounds
- Clear comparison against prior rates (PDOR, discounted-setting results).
-  Tackles the average-reward offline regime, important for continuing tasks (inventory control, admission control) where resets are impossible
- Optimal-rate guarantees were missing in this setting.

# Weaknesses
- Results hinge on linear approximation
- Pessimism paradigm already popular in episodic/dis­counted RL

---

> ### Author Rebuttal · Authors · 2025-07-31
>
> We thank the reviewer for the positive feedback.
>
> The main obstacle in directly replacing the linear critic with a neural network is the method we use to enforce pessimism. Specifically, the variable $\xi$ in (6) is constrained to a confidence ellipsoid $\lVert \xi \rVert_{\hat{\Lambda}} \leq \beta$. The parameter $\beta$ quantifies the uncertainty in the dataset. This is determined precisely using martingale concentration results from the linear MDP literature. It is not yet clear how to extend this technique to more general models such as neural networks.
>
> There are methods in the literature that implement pessimism for more general approximation. However, they need to sacrifice the precise uncertainty quantification that can be achieved in the linear setting. Two relevant papers mentioned in the related work section are those of [Xie et al., 2021] and [Cheng et al., 2022], which focus on the discounted setting. Algorithm 1 in [Xie et al., 2022] alternates between policy optimization steps and a critic which chooses a pessimistic approximation model from a general function class. When specialized to the case of linear function approximation, their algorithm can be viewed as solving a Lagrangian relaxation of a constrained problem similar to ours, but with only the linear constraint. Since they work with more general function approximation, they are not able to quantify the uncertainty in the dataset to the same level of precision as we are in our paper. Instead, the amount of pessimism is determined by the Lagrange multiplier, which is a tunable hyper-parameter. The algorithm is very general, but when specified to the linear setting, the lack of precise uncertainty quantification results in convergence rates of only $N^{-1/3}$ for $N$ data samples.
>
> It would be interesting to see if similar ideas work for average reward MDPs, but it does not seem trivial due to the presence of the additional variable $J$ in the average reward setting. We leave these interesting questions for future work.

---

> > ### Comment · Reviewer_wWmr · 2025-08-06
> >
> > Thank you for the clarification. I will maintain my positive score.

---

### Decision · Program_Chairs · 2025-09-17

**Decision:**

Accept (poster)

**Comment:**

This paper studies an offline actor-critic algorithm for policy optimization on average-reward MDPs with linear (value) function approximation using the pessimism principle. All the reviewers were positive about the contributions of the paper, despite noting some limitations (e.g. reliance on linear approximation, lack of computational efficiency of the procedure). The authors are recommended to include the minor clarifications that they provided to reviewers during the rebuttal/discussion phase in the camera-ready version.